# Akaluc/AkaLumine bioluminescence system enables highly sensitive, non-invasive and temporal monitoring of gene expression in *Drosophila*

Akira Ito[1], Nagisa Matsuda[1], Yumiko Ukita[1], Misako Okumura [1,2] & Takahiro Chihara [1,2✉]

Bioluminescence generated by luciferase and luciferin has been extensively used in biological research. However, detecting signals from deep tissues in vivo poses a challenge to traditional methods. To overcome this, the Akaluc and AkaLumine bioluminescent systems were developed, resulting in improved signal detection. We evaluate the potential of Akaluc/ AkaLumine in *Drosophila melanogaster* to establish a highly sensitive, non-invasive, and temporal detection method for gene expression. Our results show that oral administration of AkaLumine to flies expressing Akaluc provided a higher luminescence signal than Luc/D-luciferin, with no observed harmful effects on flies. The Akaluc/AkaLumine system allows for monitoring of dynamic temporal changes in gene expression. Additionally, using the Akaluc fusion gene allows for mRNA splicing monitoring. Our findings indicate that the Akaluc/ AkaLumine system is a powerful bioluminescence tool for analyzing gene expression in deep tissues and small numbers of cells in *Drosophila*.

[1] Program of Biomedical Science, Graduate School of Integrated Sciences for Life, Hiroshima University, 1-3-1 Kagamiyama, Higashi-Hiroshima, Hiroshima, Japan. [2] Program of Basic Biology, Graduate School of Integrated Sciences for Life, Hiroshima University, 1-3-1 Kagamiyama, Higashi-Hiroshima, Hiroshima, Japan. ✉email: tchihara@hiroshima-u.ac.jp

Bioluminescence is used as a reporter in a wide range of biological research, one of which is gene expression analysis[1]. Although quantitative real-time PCR (qPCR) is also used for gene expression analysis, this technique requires nucleic acid extraction from biological samples, such as cells and animals. Therefore, it is necessary to prepare multiple samples for temporal analysis. In contrast, the bioluminescence-based method is non-invasive because it analyzes gene expression from the luminescence produced by the reaction between luciferase and luciferin[2]. This characteristic enables relatively easy analysis of gene expression over time and in the same sample. Because of this advantage, bioluminescence-based gene expression analysis is used to monitor the temporal expression of circadian rhythm genes[3] and screen drugs that affect the expression of specific genes[4].

In general, bioluminescence-based methods use luciferases and luciferins derived from insects, such as firefly luciferase (Luc) and D-luciferin, or from marine organisms, such as Renilla luciferase (Rluc) and coelenterazine[5]. However, luciferases and luciferins exhibit several disadvantages in vivo. First, the wavelength of light produced by the reaction between luciferases and luciferins cannot easily penetrate animal tissues. The peak emission wavelength of Luc/D-luciferin is approximately 578 nm and that of Rluc/coelenterazine is approximately 482 nm, which is easily absorbed by hemoglobin[5,6]. As a result, signals from deep tissue in the body are attenuated. Second, the low tissue permeability of the substrate, D-luciferin, causes its heterogeneous distribution in vivo. In particular, the brain permeability is remarkably low[7]. To solve these problems, Akaluc and AkaLumine have been developed. AkaLumine is an artificially synthesized analog of D-luciferin[8]. In addition, AkaLumine-HCl was developed to improve water solubility[9]. The maximum emission wavelength produced by the reaction between Luc and AkaLumine-HCl is 677 nm in the near-infrared region, indicating high tissue permeability. Akaluc is an enzyme that was developed through directed evolution by introducing mutations in Luc to produce brighter luminescence in the reaction with AkaLumine[10]. Using Akaluc and AkaLumine-HCl, the detection of luminescent signals from deep tissues, particularly from the brain, in mice and marmosets has dramatically improved.

In Drosophila research, Firefly luciferase and D-luciferin are commonly used. The Drosophila head is about 1 mm thick and is not "deep" compared to the brains of mammals, but it is known that photons are absorbed or scattered by pigments in the eyes and cuticle of Drosophila[11] (Supplementary Fig. 1). Therefore, the use of Akaluc/AkaLumine, which produces a signal with better tissue permeability than Luc/D-luciferin, would improve the analysis using bioluminescence in Drosophila.

In this study, we introduced Akaluc/AkaLumine into Drosophila melanogaster to establish a highly sensitive, non-invasive, and continuous gene expression analysis. We generated transgenic strains expressing Akaluc under the control of the GAL4/UAS system and knock-in strains expressing Akaluc from endogenous promoters. Using these strains, we investigated whether the Akaluc/AkaLumine system is superior to Luc/D-luciferin, which is widely used in Drosophila research, and whether this system can be used to perform non-invasive, continuous gene expression analysis in Drosophila. First, we examined the conditions and toxicity of oral administration of AkaLumine, a substrate of Akaluc, to Drosophila. Next, the luminescence levels of Akaluc/AkaLumine and Luc/D-luciferin were compared. The amount of signal detected in the nervous system was significantly higher with Akaluc/AkaLumine than with Luc/D-luciferin. Furthermore, we provided evidence that the Akaluc/AkaLumine system is useful for monitoring the temporal dynamics of genes of interest in vivo. We showed that Akaluc/AkaLumine can be used

in Drosophila to perform temporal gene expression analyses with higher sensitivity than the traditional Luc/D-luciferin system.

## Results

### Oral administration of AkaLumine to *Drosophila* expressing Akaluc enables in vivo detection of luminescence signals with a high signal-to-noise ratio.

Intravenous, intraperitoneal, and oral administration of AkaLumine has been tested in mice[10]. We administered AkaLumine orally for easy and intact substrate delivery to Drosophila. AkaLumine-HCl (AkaLumine) was used in all the experiments in this study. We added a single fly and food containing AkaLumine to each well of a 24-well plate and measured the luminescence (Fig. 1a). This method allows continuous luminescence measurements while feeding AkaLumine to freely moving flies. Flies expressing Akaluc ubiquitously showed significantly higher luminescence after oral administration of AkaLumine than the background signal exhibited by flies without AkaLumine (Fig. 1b). In addition, a negligible luminescent signal was detected when AkaLumine was administered to wild-type flies, indicating that AkaLumine is unlikely to produce luminescence by oxidation with the endogenous enzyme present in Drosophila. These results indicate that the oral administration of AkaLumine to Akaluc flies allows for luminescence detection with a high signal-to-noise ratio. Next, we determined the appropriate AkaLumine concentration for luminescence production by Drosophila. When flies ubiquitously expressing Akaluc (*tubP-Gal4 > UAS-Venus-Akaluc*) were administered AkaLumine, the luminescent signal increased with increasing AkaLumine concentrations (Fig. 1c). The luminescence peaked at ~1.0 mM of AkaLumine. We also investigated the temporal changes in the luminescence signals after AkaLumine administration or AkaLumine withdrawal. At concentrations of 0.5, 1.0, and 2.0 mM, the luminescent signal increased quickly and peaked at approximately 5 h after AkaLumine administration (Fig. 1d). We also found that the higher the concentration of AkaLumine administered, the slower the rate of luminescent signal decrease after stopping AkaLumine administration (time required for mean counts per second (cps) to fall below 6,000 after stopping AkaLumine administration, 0.1 mM: 2.6 h, 0.5 mM: 6.2 h, 1.0 mM: 8.8 h, 2.0 mM: 22.8 h). In addition, we compared the changes in luminescence levels of individual flies until 48 h after feeding 1.0 mM AkaLumine. We found no notable differences, and the signal was stable in all individuals (Supplementary Fig. 2). Taken together, these results indicate that the oral administration of AkaLumine at appropriate concentrations to Akaluc-expressing Drosophila enables highly sensitive in vivo detection of bioluminescence.

### Oral administration of AkaLumine is not toxic to *Drosophila*.

We investigated whether AkaLumine administration was detrimental to Drosophila survival. We fed AkaLumine (1.0 mM) to flies for 24 h and measured the expression levels of several stress-response genes. *Glutathione S transferase D1* (*GstD1*) expression is upregulated by oxidative stress and aging[12]. Endoplasmic reticulum (ER) stress induces the expression of *Bip*, an ER chaperone[13]. *Atg8a*, a key component of autophagy, is upregulated under starvation stress[14]. Expression of the apoptosis activator *head involution defective* (*Hid*), is induced by cell death signals[15]. The expression levels of these genes were not affected by AkaLumine treatment (Fig. 2a, normalized by *gapdh2* gene). Similar results were obtained when gene expression levels were normalized to the reference gene, *actin 5c* (Fig. 2b). These results indicate that the oral administration of AkaLumine does not cause detrimental stress in flies.

We investigated the effects of long-term administration of AkaLumine. Fly eggs were placed on fly food containing

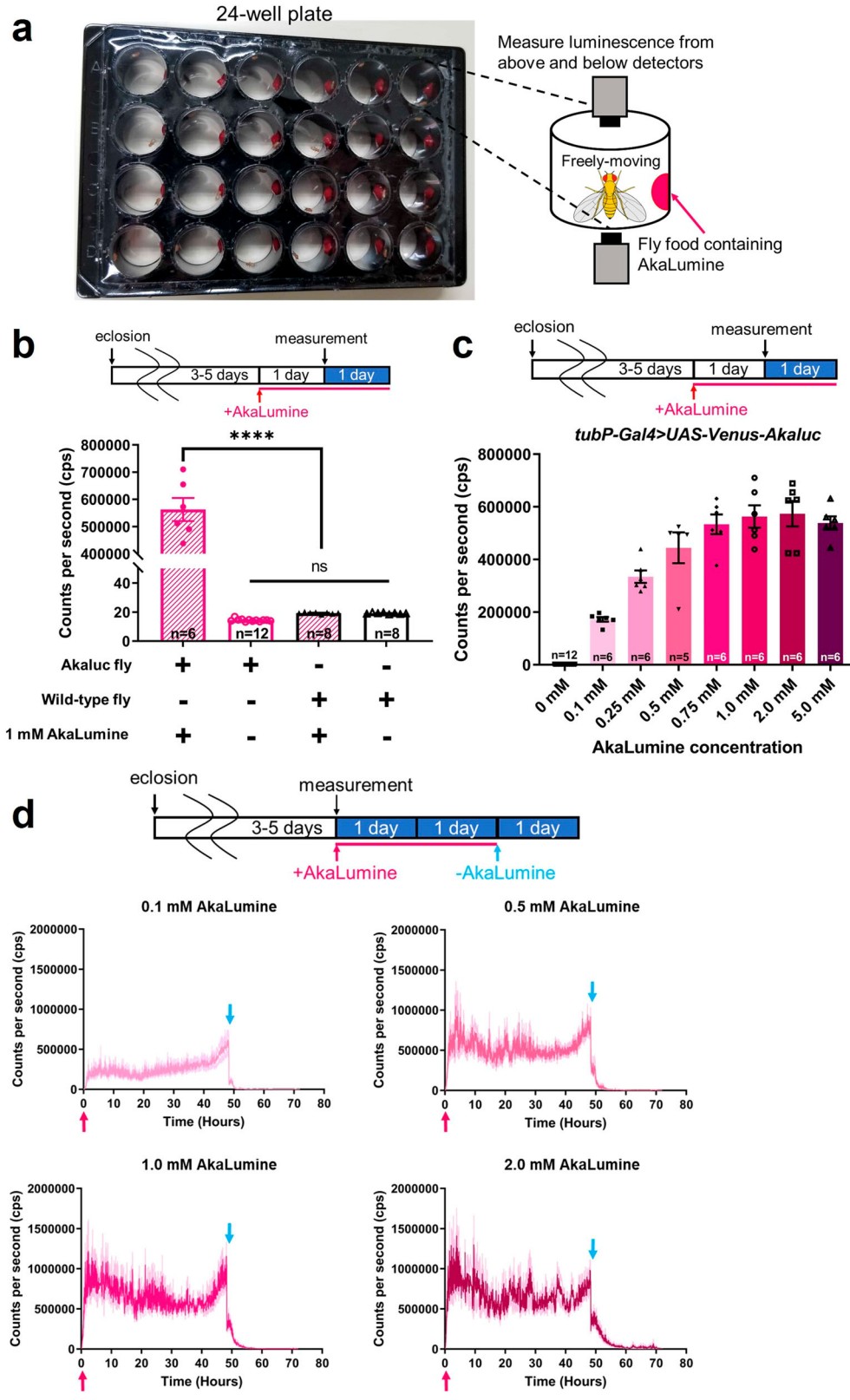

AkaLumine (1.0 mM) and reared until adulthood. Rearing flies on AkaLumine food did not significantly affect pupariation or eclosion (Fig. 2c, d). These results indicate that long-term AkaLumine administration did not affect fly development. Together, these results suggest that oral administration of AkaLumine has little or no toxicity in *Drosophila*.

**Akaluc/AkaLumine emits stronger bioluminescence signals from a small number of neurons than Luc/D-luciferin.** In mammals, the use of Akaluc/AkaLumine dramatically increases the strength of signals from deep tissues in vivo compared with conventional Luc/D-luciferin[10]. We compared the luminescence signals of Akaluc/AkaLumine and Luc/D-luciferin in *Drosophila*.

**Fig. 1 Detection of Akaluc/AkaLumine bioluminescence by oral administration of AkaLumine in *Drosophila*. a** Schematic diagram of the luminescence measurements. Each well of the 24-well plate contained a single fly and food containing AkaLumine. The luminescence emitted from the fly freely moving in the well was measured by the upper and lower bioluminescence detectors. **b** Luminescence was detected by administering AkaLumine to flies expressing Akaluc. Akaluc flies (*tubP-Gal4 > UAS-Venus-Akaluc*) or wild-type flies (*w1118*) were used. In AkaLumine administration groups, flies were fed food with AkaLumine (1.0 mM) for 24 h prior to the start of luminescence measurements, and measurements were performed continuously for 24 h. The group not administered AkaLumine was fed a normal diet and measurements were taken. The graph shows the mean value of luminescence measured continuously for 24 h in each trial. The number of trials for each data set is shown on the graph. One-way ANOVA with Tukey-Kramer test was used for a statistical test. ****$p < 0.0001$, ns no significance. Error bars indicate s.e.m. **c** The examination of appropriate AkaLumine concentrations for in vivo monitoring of Akaluc/ AkaLumine bioluminescence. Akaluc flies (*tubP-Gal4 > UAS-Venus-Akaluc*) were orally administered each concentration of AkaLumine 24 h prior to the start of luminescence measurements, and measurements were performed continuously for 24 h. The group with zero AkaLumine was fed normal food. The graph shows the mean value of luminescence measured continuously for 24 h in each trial. The number of trials for each data set is shown in the graph. Error bars indicate s.e.m. **d** The temporal changes of bioluminescence after the administration or withdrawal of AkaLumine food. Luminescence measurements were started as soon as AkaLumine was administered to Akaluc-expressing flies (red arrows). At 48 h after the start of measurements, the flies were transferred to wells with normal food without AkaLumine under cold anesthesia (blue arrows), and bioluminescence measurements were carried out for an additional 24 h. The darker colored line in each graph indicates the mean and the lighter colored line indicates the error bars (s.e.m). The number of trials for each graph is $n = 6$.

When we measured the luminescence level in flies ubiquitously expressing Luc (*tubP-Gal4 > UAS-Luc*), the luminescence level increased as the concentration of D-luciferin increased (Fig. 3a). At low substrate concentrations, the Akaluc/AkaLumine signal tended to be stronger than that of Luc/D-luciferin (Figs. 1c, 3a, Akaluc + 0.5 mM AkaLumine mean cps: $4.4 \times 10^5$ cps, Luc + 0.5 mM D-luciferin mean cps: $2.7 \times 10^5$ cps). In contrast, at high substrate concentrations, the Luc/D-luciferin signal was stronger than that of Akaluc/AkaLumine (Akaluc + 5.0 mM AkaLumine mean cps: $5.3 \times 10^5$ cps, Luc + 5.0 mM D-luciferin mean cps: $1.0 \times 10^6$ cps). These results suggest that the reaction between Akaluc and AkaLumine is more efficient and emits higher luminescence even at low AkaLumine concentrations; however, the luminescence of Luc/D-luciferin is stronger than that of Akaluc/AkaLumine when luciferase is expressed in the whole body.

We examined the luminescence signals of Akaluc/AkaLumine and Luc/D-luciferin in specific tissues. When the pan-neuronal driver *elav-Gal4* was used to express Akaluc or Luc, the Akaluc/ AkaLumine signal was significantly higher than that of Luc/D-luciferin at both low and high substrate concentrations (Fig. 3b). *OK107-Gal4* is mainly expressed in the mushroom body which is composed of ~2500 Kenyon cells[16]. The Akaluc/AkaLumine signal was significantly stronger than that of Luc/D-luciferin when *OK107-Gal4* was used to express Akaluc or Luc (Fig. 3c). Or42b is one of the olfactory receptors (Ors), and is expressed in about 50–90 olfactory receptor neurons (ORN)[16]. When luminescent signals from Or42b ORNs were detected, Akaluc/ AkaLumine provided a stronger signal than Luc/D-luciferin (Fig. 3d). These results indicate that Akaluc/AkaLumine is useful for detecting luminescent signals from deep tissues, such as the nervous system, and a small number of cells, such as Or42b-expressing ORNs, in *Drosophila*.

Or85a is also an Or, and the number of ORNs expressing Or85a is ~20–50[17]. We compared the luminescence signals between flies expressing Akaluc in Or42b (*Or42b-Gal4 > UAS-Venus-Akaluc*) and Or85a (*Or85a-Gal4 > UAS-Venus-Akaluc*) ORNs and found that a higher signal was detected in flies expressing Akaluc in Or42b ORNs than in those expressing Or85a ORNs (Fig. 3e). However, when Luc/D-luciferin was used, the difference in Or42b and Or85a cell numbers could not be detected by luminescence. These results indicate that a small difference in cell number can be detected as a difference in the luminescence signal using Akaluc/AkaLumine.

**Akaluc/AkaLumine enables bioluminescence imaging in the *Drosophila* brain.** In mice and marmosets, the use of Akaluc/

AkaLumine dramatically improves imaging based on luminescent signals, particularly in deep tissues[10]. We investigated whether Akaluc/AkaLumine could improve bioluminescence imaging in *Drosophila*. When Akaluc or Luc was ubiquitously expressed, Akaluc/AkaLumine was sufficient for imaging at lower concentrations than Luc/D-luciferin (Fig. 4a–c). Interestingly, Akaluc was successfully imaged in the brain and ventral nerve cords when expressed in the nervous system (Fig. 4d–f). In contrast, almost no signal was detected when Luc was used. These results imply that the signals detected in Luc-expressing flies by *tubP-Gal4* are likely derived from body surface tissues and not from the nervous system. In addition, we tried to image the intestinal shape from Akaluc/AkaLumine signals using the intestine-specific Gal4 driver (*mex1-Gal4 > UAS-Venus-Akaluc*). We could not determine the shape of the gut from the signal, but we were able to image the signal from only the abdomen of *Drosophila* (Supplementary Fig. 3). These results suggest that Akaluc/AkaLumine is suitable for bioluminescence imaging in deep tissues in *Drosophila*, with spatial resolution at the level of body parts such as the head or abdomen.

**Akaluc/AkaLumine provides highly sensitive and temporal gene expression analysis.** Since gene expression analysis using bioluminescence is a non-invasive method, it is relatively easy to perform continuous measurements using the same individual[2]. We investigated whether Akaluc/AkaLumine can be used to monitor gene expression non-invasively and temporally in *Drosophila*. For this purpose, we generated a *Drosophila* strain that expressed *Akaluc* in an expression-dependent manner on the innate immunity-related gene *Induced by Infection* (*IBIN*). In this strain, the *Akaluc* gene was knocked-in at the *IBIN* locus (*IBIN^Akaluc-KI*) (Fig. 5a). *IBIN* is an innate immune-related gene whose expression is greatly increased by infection with Gram-negative or Gram-positive bacteria and is involved in promoting bacterial resistance[18]. We expected that *IBIN^Akaluc-KI* flies would produce a luminescent signal upon infection, along with an increase in *IBIN* expression. As expected, we were able to monitor a large increase in luminescent signal in the *E. coli* DH5α-infected group compared to the DH5α-uninfected group (Fig. 5b). To further expand the utility of this innate immune monitoring system, we generated a transgenic *IBINp-Venus-Akaluc* strain expressing *Akaluc* under the control of a putative *IBIN* promoter region (319 bp between *CG30109* and *IBIN*, Fig. 5a). When the *IBINp-Venus-Akaluc* strain was infected with DH5α, an increased luminescent signal was observed in a similar pattern to that of *IBIN^Akaluc-KI* (Fig. 5c). Next, we compared temporal changes in luminescent signals from *IBIN^Akaluc-KI* and *IBINp-Venus-Akaluc*

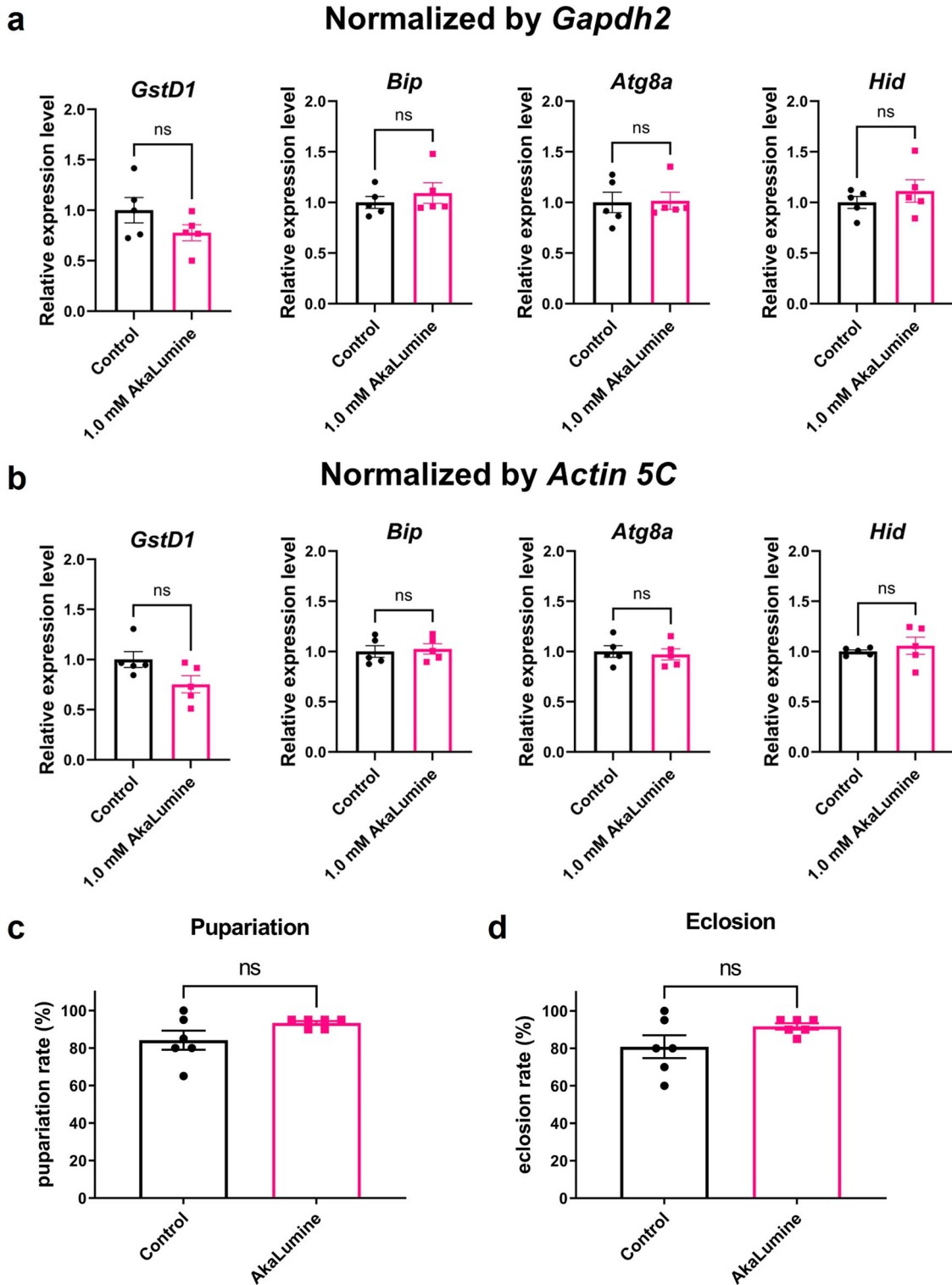

**Fig. 2 The effects of AkaLumine administration on the stress response and developmental defects of *Drosophila*. a, b** Short-term oral administration of AkaLumine to flies did not induce oxidative stress, ER stress, starvation stress, or cell death responses. Gene expression in wild-type flies ($w^{1118}$) fed with food containing AkaLumine (1.0 mM) for 24 h was measured by qPCR. As a control, food without AkaLumine was used. Expression levels were normalized using *Gapdh2* (**a**) and *Actin 5 C* (**b**) as an internal control gene. An unpaired *t*-test was used for statistical analysis. ns: no significance, $n = 5$. Error bars indicate s.e.m. **c, d** Long-term administration of AkaLumine did not affect the development of flies. There was no significant difference in the pupation rate (**c**) and eclosion rate (**d**) when the wild-type flies ($w^{1118}$) were raised from embryo to adult on fly food with or without AkaLumine. An unpaired *t*-test was used for statistical analysis. ns no significance, $n = 6$. Error bars indicate s.e.m.

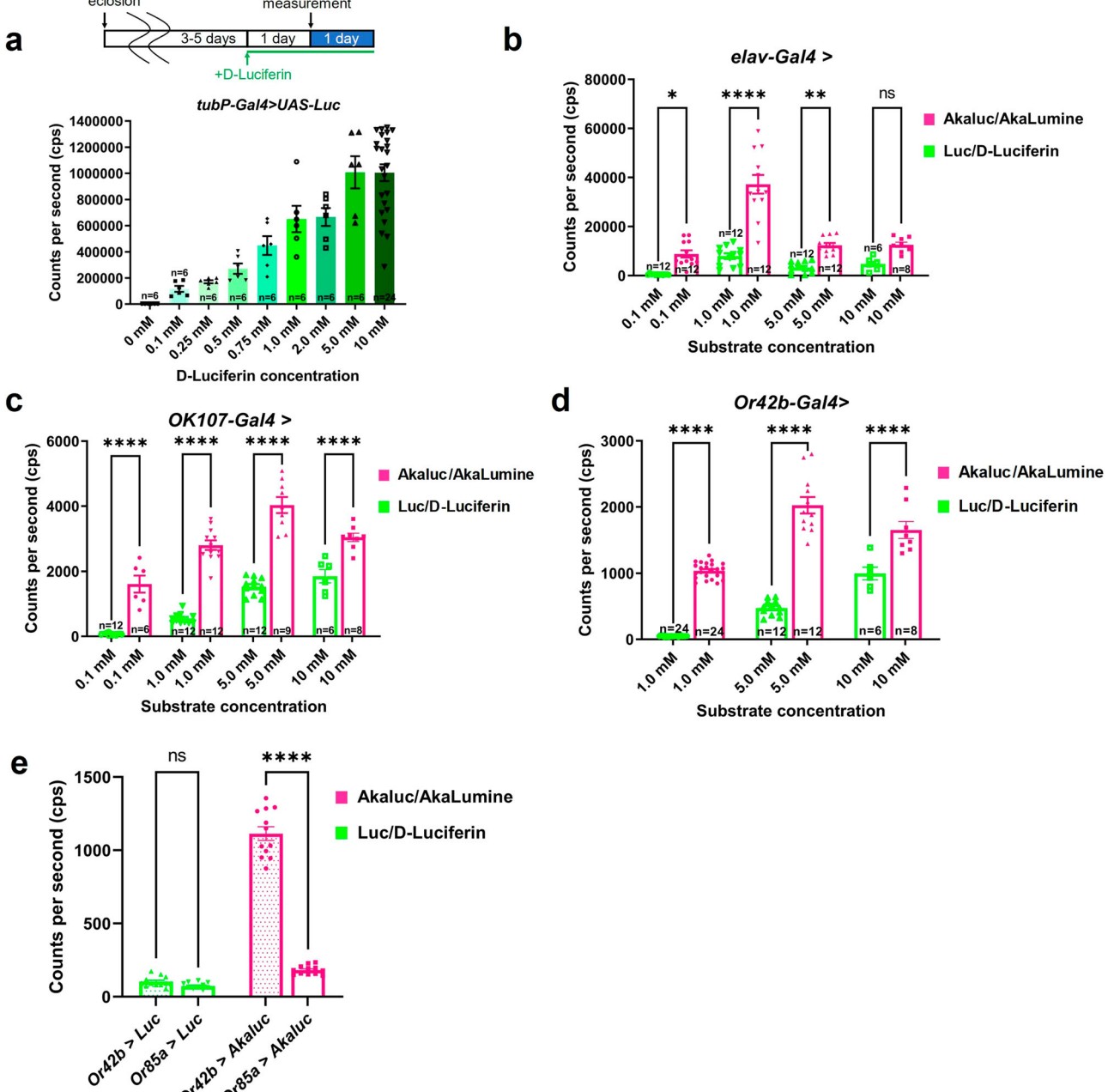

**Fig. 3 Bioluminescent signals generated from Akaluc/AkaLumine and Luc/D-luciferin in deep tissues and cells. a** Flies expressing Luc ubiquitously (*tubP-Gal4 > UAS-Luc*) were orally administered each concentration of D-luciferin 24 h before the start of bioluminescence measurement. All of the experiments in Fig. 3 were performed using the same method of substrate administration (AkaLumine or D-luciferin) and luminescence measurement. The group with zero D-luciferin was fed normal food. The graph shows the mean value of luminescence measured continuously for 24 h in each trial. The number of trials for each data set is shown in the graph. Error bars indicate s.e.m. **b, c** The bioluminescence of Akaluc/AkaLumine detected in the nervous system was stronger than that of Luc/D-luciferin. Flies expressing Akaluc or Luc by *elav-Gal4* (b: *elav-Gal4 > UAS-Venus-Akaluc* and *elav-Gal4 > UAS-Luc*) and *OK107-Gal4* (**c**: *OK107-Gal4 > UAS-Venus-Akaluc* and *OK107-Gal4 > UAS-Luc*) were administered each concentration of AkaLumine or D-luciferin, respectively. One-way ANOVA with Tukey–Kramer test was used for statistical analysis. *$p < 0.05$, **$p < 0.01$, ****$p < 0.0001$, ns no significance. The number of trials for each data set is shown in the graphs. Error bars indicate s.e.m. **d** The bioluminescence signal of Akaluc/AkaLumine detected in the Or42b ORN was higher than that of Luc/D-Luciferin. Flies expressing Akaluc or Luc by Or42b-Gal4 (*Or42b-Gal4 > UAS-Venus-Akaluc* and *Or42b-Gal4 > UAS-Luc*) were fed AkaLumine (1.0, 5.0, 10 mM) or D-luciferin (1.0, 5.0, 10 mM), respectively. One-way ANOVA with Tukey–Kramer test was used for statistical analysis, ****$p < 0.0001$. The number of trials for each data set is shown in the graphs. Error bars indicate s.e.m. **e** Flies expressing Akaluc by *Or42b-Gal4* (*Or42b-Gal4 > UAS-Venus-Akaluc*) or *Or85a-Gal4* (*Or85a-Gal4 > UAS-Venus-Akaluc*) were administered AkaLumine (1.0 mM) and the luminescence levels were compared. Flies expressing Luc by *Or42b-Gal4* (*Or42b-Gal4 > UAS-Luc*) or *Or85a-Gal4* (*Or85a-Gal4 > UAS-Luc*) were administered D-luciferin (5.0 mM) and the luminescence levels were compared. One-way ANOVA with Tukey–Kramer test was used for statistical analysis, ****$p < 0.0001$, ns: no significance, $n = 12$. Error bars indicate s.e.m.

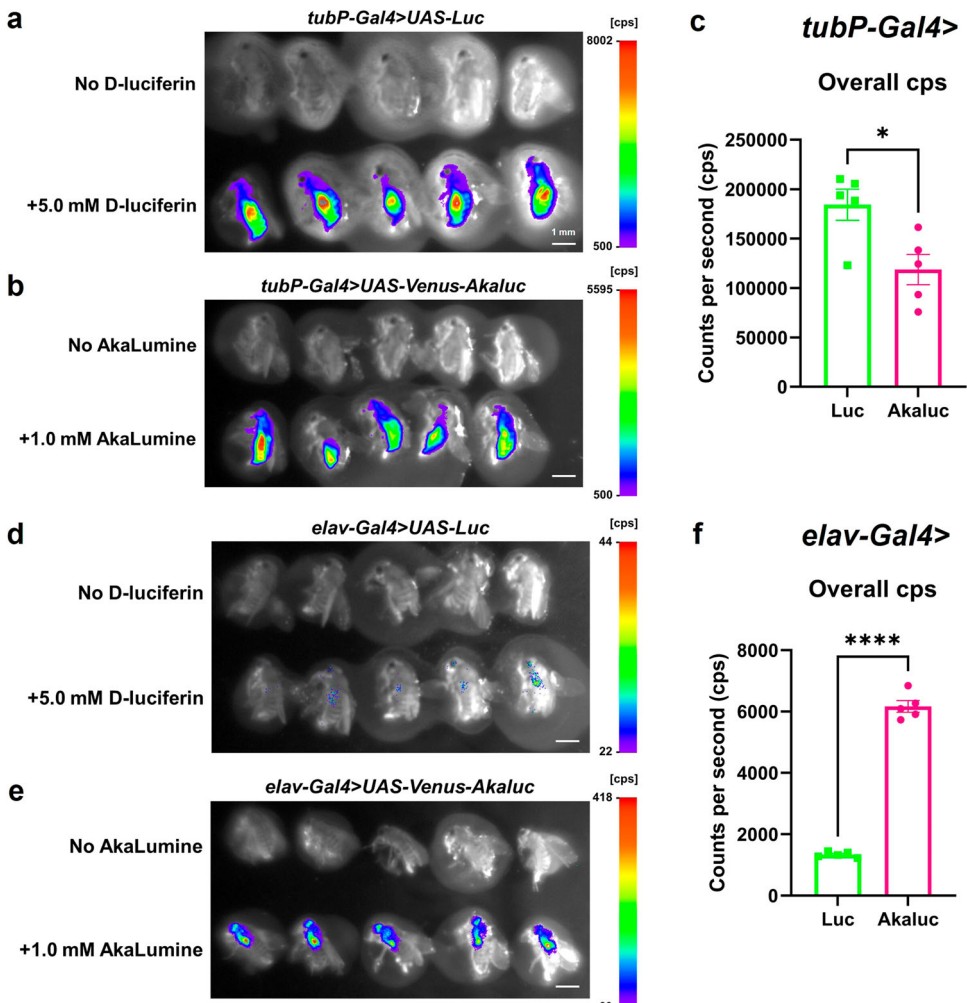

**Fig. 4 Akaluc/AkaLumine bioluminescence imaging in *Drosophila*. a, b** *tubP-Gal4 > UAS-Luc* (**a**) and *tubP-Gal4 > UAS-Venus-Akaluc* (**b**) flies are shown. D-luciferin (5.0 mM) or AkaLumine (1.0 mM) was administered for 24 h to Luc flies or Akaluc flies, respectively (lower flies). The control group was fed normal food (upper flies). Scale bar at the bottom right of each photo indicates 1 mm. **c** From the imaging results, the overall counts per second (cps) of Luc/D-luciferin and Akaluc/AkaLumine was compared. Unpaired *t*-test was used for statistical analysis. *$p < 0.05$, $n = 5$. Error bars indicate s.e.m. **d, e** *elav-Gal4 > UAS-Luc* (**d**) and *elav-Gal4 > UAS-Venus-Akaluc* (**e**) flies are shown. D-luciferin (5.0 mM) or AkaLumine (1.0 mM) was administered for 24 h to Luc flies or Akaluc flies, respectively (lower flies). The control group was fed normal food (upper flies). Scale bar at the bottom right of each photo indicates 1 mm. **f** From the imaging results, the overall cps of Luc/D-luciferin and Akaluc/AkaLumine was compared. Unpaired *t*-test was used for statistical analysis. ****$p < 0.0001$, $n = 5$. Error bars indicate s.e.m.

strains with actual *IBIN* expression by qPCR. The qPCR results showed that *IBIN* expression decreased 24 h after infection (Fig. 5d), whereas the results using *IBIN^Akaluc-KI^* or *IBINp-Venus-Akaluc* showed high luminescence levels for over 24 h (Fig. 5b, c). This was probably due to the slow degradation rate of Akaluc. Therefore, we generated flies with the PEST sequence connected to the Akaluc sequence (*IBINp-Venus-Akaluc-PEST*). PEST sequences are rich in proline (P), glutamate (E), serine (S), and threonine (T), which are found in proteins with short lifespans[19,20]. The addition of this PEST sequence to the Luciferase shortens the intracellular lifetime of the Luciferase protein and allows for monitoring with improved time resolution[21]. Using *IBINp-Venus-Akaluc-PEST*, we found that the luminescent signals increased with bacterial infection and decreased 24 h after infection, similar to the qPCR results (Fig. 5e). These results indicate that temporal gene expression analysis can be performed on the same individual in *Drosophila* using Akaluc/AkaLumine and that the addition of the PEST sequence to Akaluc successfully improved the time resolution.

**Fusion of Akaluc to endogenous protein may affect protein function.** To monitor the endogenous levels of the protein of interest, we generated *Akaluc* knocked-in allele. The *Bruchpilot* (*Brp*) protein is localized in the active zone of the presynapse, and Brp protein levels are significantly increased by sleep deprivation[22,23]. We inserted *Akaluc* into the 3′-end of the *brp* gene to generate a *brp-Akaluc* strain (*brp^Akaluc-KI^*) which produces Brp-Akaluc fusion protein (Supplementary Fig. 4a). Using this *brp^Akaluc-KI^* strain, we investigated whether sleep-deprivation-induced Brp increases could be monitored. Unexpectedly, 24 h sleep deprivation could not increase Brp protein levels in the *brp^Akaluc-KI^* strain, which were detected by luminescence monitoring and Western blotting analysis (Supplementary Fig. 4b–e). On the other hand, when we used wild-type flies, we observed an increase in Brp protein levels with sleep deprivation, consistent with previous studies, indicating that sleep deprivation was successfully achieved with our technique (Supplementary Fig. 4f). These results suggest that the fusion of Akaluc may have affected the functions of Brp, such as transport

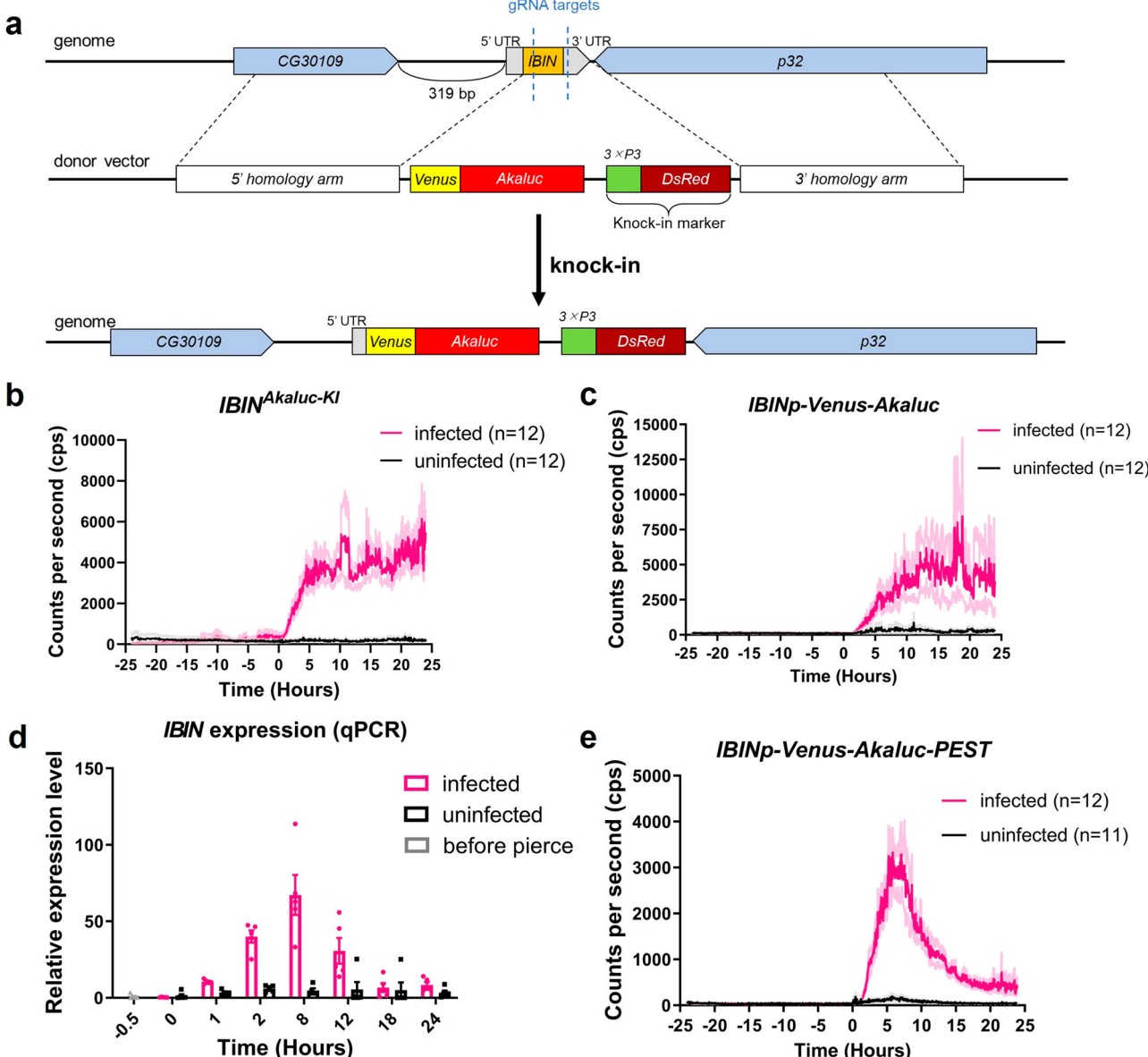

**Fig. 5 The monitoring the temporal expression changes of innate immunity-related gene, IBIN.** **a** Schematic diagram of *IBIN*$^{Akaluc-KI}$ strain generation. The coding region and 3′ UTR of *IBIN* were replaced with *Venus-Akaluc* and the knock-in marker *DsRed* by homologous recombination using the CRISPR/ Cas9 system. **b, c, e** *IBIN*$^{Akaluc-KI}$ (**b**), *IBINp-Venus-Akaluc* (**c**), and *IBINp-Venus-Akaluc-PEST* (**e**) strains could monitor the increase of *IBIN* expression induced by bacterial infection. Bioluminescence measurements were started 24 h prior to infecting the flies with the bacteria (−24 h timepoint). The flies were infected with *E. coli* (DH5α) (magenta line, 0 h timepoint), and measurements were performed for another 24 h. The "uninfected group" was pricked by a needle without bacteria (black line). AkaLumine was administered to the flies 24 h before the luminescence measurements were started, and the flies were constantly fed AkaLumine during the measurements. The darker colored line indicates the mean and the lighter colored line indicates the error bars (s.e.m). The number of trials is shown in the graph. **d** Measurement of *IBIN* expression using qPCR. At time point 0 h, the "infected" flies were infected with DH5α (magenta) and the "uninfected" flies were pricked by a needle without bacteria (black). At the −0.5 h time point, real-time PCR was performed using intact flies before needle piercing (gray). RpL32 was used as an internal control. $n = 5$, error bars indicate s.e.m.

and localization, and that this should be carefully considered when fusing Akaluc to the protein of interest.

**Akaluc/AkaLumine to measure ER stress levels over time.** We investigated the potential use of Akaluc/AkaLumine to monitor temporal changes in protein levels in vivo, we used *X-box binding protein 1* (*Xbp1*), a transcription factor that acts in the *Inositol-requiring enzyme 1* (*Ire1*) pathway, an unfolded protein response (UPR) signaling pathway. XBP1 shows no transcription factor activity under normal conditions. However, under ER stress, XBP1 becomes an active transcription factor through unconventional

splicing induced by IRE1 and leads to the expression of UPR target genes[24]. Using this property of *Xbp1*, ER stress reporters expressing activated XBP1 fused with GFP under ER stress have been developed[25,26]. Similarly, we aimed to generate an ER stress reporter using the Akaluc/AkaLumine system. We generated an *Xbp1-Akaluc* strain (*UAS-Xbp1-Venus-Akaluc*) by linking Akaluc in-frame to the activated XBP1 protein generated by unconventional splicing (Fig. 6a). Heat shock causes protein misfolding and unfolding, and induces unconventional splicing of *Xbp1*[27,28]. In flies ubiquitously expressing *Xbp1-Akaluc* (*tubP-Gal4 > UAS-Xbp1-Venus-Akaluc*), a significant increase in the bioluminescence signal

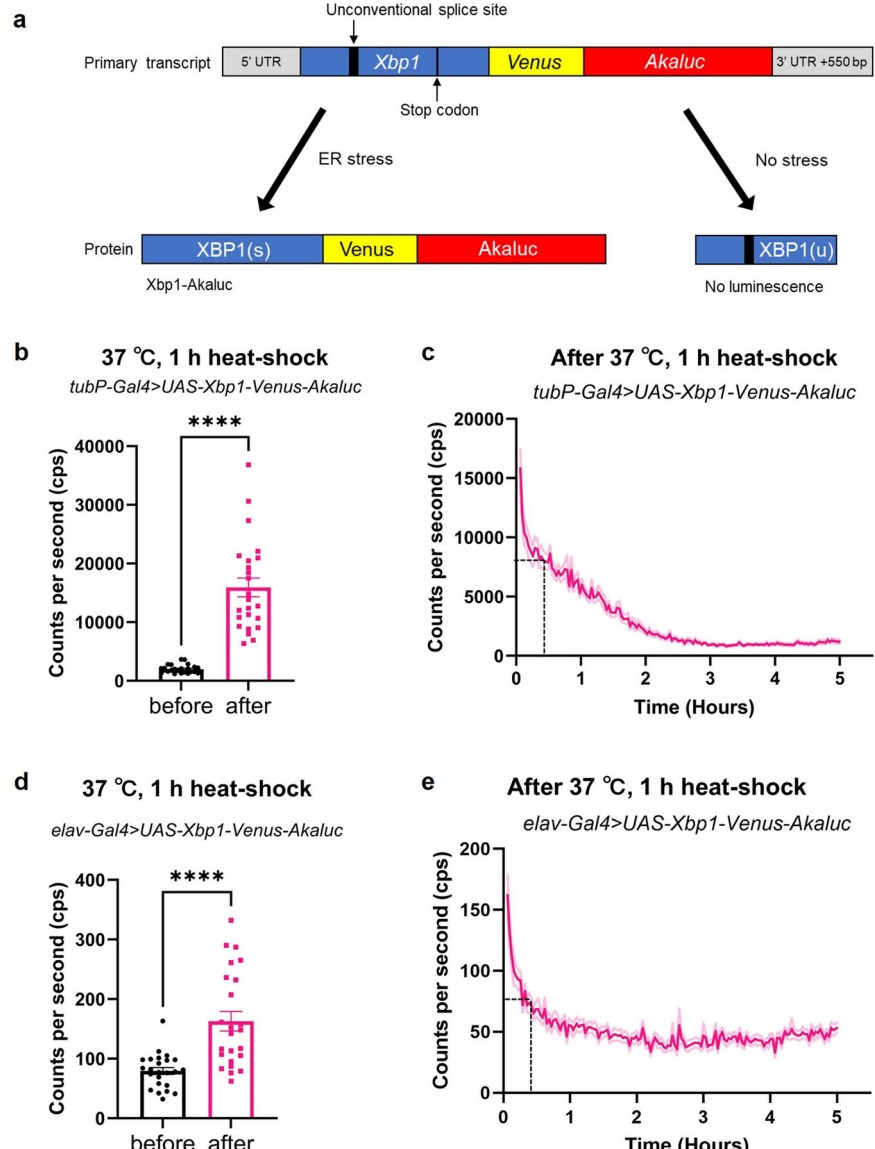

**Fig. 6 The monitoring of the temporal changes in ER stress induced by heat shock using *Xbp1-Akaluc*. a** Schematic diagram indicating the unconventional splicing of *Xbp1-Venus-Akaluc* transcript. Under the ER stress condition, a frameshift was induced by unconventional splicing, resulting in the loss of the stop codon. Thus, the transcription factor XBP1(s) fused with Venus-Akaluc is generated (left). Under normal conditions (no stress), translation is terminated in the middle of *Xbp1* transcript by a stop codon. Thus, the following sequences including *Venus-Akaluc* are not translated. This produces only XBP1(u), which has no activity as a transcription factor (right). **b**–**e** Heat shock-induced changes in the expression of activated XBP1 in the whole body and nervous system could be monitored by *Xbp1-Akaluc*. tubP-Gal4 > UAS-Xbp1-Venus-Akaluc (**b**, **c**) and *elav-Gal4 > UAS-Xbp1-Venus-Akaluc* (**d**, **e**) were exposed to a 1 h heat shock at 37 °C and continuous luminescence measurements were performed for 5 h. Flies were administered AkaLumine (1.0 mM) for 24 h prior to heat shock and continued to be administered during heat shock and luminescence measurements. **b** and **d** compare the amount of luminescence of the same flies before and immediately after heat shock. The dotted lines on the (**c**) and (**e**) indicate the half-life of each signal (~20–30 min). The darker colored lines in **c** and **e** indicate the mean and the lighter colored lines indicate the error bars (s.e.m). Unpaired *t*-test was used for statistical analysis, ****$p < 0.0001$, $n = 24$.

due to heat shock was observed (Fig. 6b). We were also able to monitor the rapid decrease in activated XBP1 expression after stopping the heat shock by measuring the luminescent signal over time (Fig. 6c). To investigate whether *Xbp1* unconventional splicing could be detected in deep tissues, we expressed *Xbp1-Akaluc* in the pan-neurons (*elav-Gal4 > UAS-Xbp1-Venus-Akaluc*). As a result, we were also able to monitor the significant increase in the luminescent signal by heat shock and the recovery process from post-heat shock to the steady state (Fig. 6d, e). In summary, *Xbp1-Akaluc* can be used to monitor temporal changes in ER stress levels in specific tissues of a single fly.

## Discussion

In this study, we introduced Akaluc/AkaLumine into *Drosophila* and demonstrated its usefulness for gene and protein expression analyses. After orally administering AkaLumine to Akaluc-expressing flies, we detected luminescence with a high signal-to-noise ratio (Fig. 1). Short- or long-term administration of AkaLumine to *Drosophila* had no detrimental effects on the flies (Fig. 2). We found that the signal intensity was significantly higher with Akaluc/AkaLumine than with Luc/D-luciferin when the signal was detected in deep tissues, such as the nervous system, or in a small number of cells, such as olfactory receptor

neurons (Fig. 3). Akaluc/AkaLumine was also superior to Luc/D-luciferin in the bioluminescence imaging of deep tissues in *Drosophila* (Fig. 4). We also succeeded in monitoring the changes in the expression of *IBIN*, which was greatly increased by bacterial infection (Fig. 5). Using *Drosophila* expressing Akaluc-fused XBP1, we were able to monitor changes in ER stress under heat stress condition (Fig. 6). These results suggest that Akaluc/AkaLumine is a powerful tool for gene and protein expression analysis in *Drosophila*, especially for temporal and non-invasive analysis of deep tissues and a small number of cells.

When Akaluc was expressed using *tubP-Gal4*, the signal peaked at 1.0 mM AkaLumine (Fig. 1c). In contrast, when Luc was expressed by *tubP-Gal4*, the signal peaked at 5.0 mM D-luciferin (Fig. 3a). This may be attributed to the higher affinity between the enzyme and the substrate for Akaluc/AkaLumine than for Luc/D-luciferin. In fact, it has been reported that the affinity of AkaLumine for Luc is significantly higher than that of Luciferin[9]. Therefore, Akaluc/AkaLumine was expected to reach peak signals at lower substrate concentrations than Luc/D-luciferin. However, the required AkaLumine concentration may be higher when Akaluc is expressed in specific tissues or cell types. In addition, an inhibitory by-product of the Luc/D-luciferin reaction is generated when an excessive concentration of the substrate is administered[29]. It is unclear whether the same phenomenon occurs during the Akaluc/AkaLumine reaction. Therefore, the concentration of AkaLumine administered to flies should be carefully considered, depending on the tissue in which Akaluc is expressed.

To measure luminescent signals simply and non-invasively, we orally administered AkaLumine to *Drosophila* by mixing it with fly food. Thus, AkaLumine intake depends on the feeding behavior of the flies, which might be a disadvantage of this method. Therefore, it may be difficult to apply the methods of this study to pupae that are not fed or to experiments that require fasting. A study in mice reported that luminescence was higher when AkaLumine was administered intraperitoneally than when it was administered orally[10]. If oral administration is experimentally difficult or does not provide a sufficient signal, intraperitoneal administration may solve these problems. Nevertheless, our results showed that, in many cases, oral administration of AkaLumine is sufficient to detect the signal over time in intact flies.

Using strains expressing Akaluc under the control of the *IBIN* promoter, we monitored temporal changes in *IBIN* expression during bacterial infection. However, the qPCR results showed a decrease in *IBIN* expression 24 h after *E. coli* infection, whereas *IBIN-Akaluc* maintained its luminescence signals 24 h after infection (Fig. 5b–d). This time difference between the actual gene expression and the luminescent signal was caused by the slow degradation speed of the Akaluc protein. In fact, the half-life of Akaluc protein in cultured human cells has been reported to be ~9 h[10]. To improve the resolution of temporal gene expression analysis, we added the PEST sequence to the Akaluc protein to shorten its degradation speed. Using *IBINp-Akaluc-PEST*, we were able to monitor the temporal changes in *IBIN* expression, which were more similar to the results obtained by qPCR (Fig. 5e). Therefore, *Akaluc-PEST* is particularly useful for monitoring genes with rapidly increasing or decreasing expression.

In HeLa cells, it has been reported that heat shock increases the number of unfolded proteins, which are rapidly reduced after heat shock is stopped[27]. A similar phenomenon was observed using *Xbp1-Akaluc* strain. Heat shock remarkably increased the signal of *Xbp1-Akaluc*. The luminescence level quickly decreased to the basal level after the heat shock was stopped (signal half-life was ~20–30 min, Fig. 6c). The half-life of Akaluc is ~9 h[10], whereas that of XBP1 is much shorter (<30 min)[30,31]. The rapid

decrease in the luminescence signal from the *Xbp1-Akaluc* strain after heat shock suggests that the Xbp1-Akaluc fusion protein likely reflects the endogenous degradation rate of XBP1. Furthermore, we monitored the increased expression of activated XBP1 in the nervous system induced by heat shock (Fig. 6d, e). Thus, *Xbp1-Akaluc* is a useful tool for temporal monitoring of ER stress in *Drosophila*.

Our results show that Akaluc/AkaLumine is particularly useful for analysis in deep tissues such as nervous system and in a small number of cells, with up to about 5-fold greater signal than Luc/D-luciferin, which is commonly used in *Drosophila* studies (Fig. 3b; Akaluc + 1.0 mM AkaLumine vs. Luc + 1.0 mM D-luciferin), and similar results have been shown in mice and marmosets[10,32]. It has been reported that Luc/D-luciferin can produce a stronger signal than Akaluc/AkaLumine by increasing the substrate concentration administered[33]. Consistent with this, in *Drosophila*, Luc/D-luciferin showed a stronger signal than Akaluc/AkaLumine under high substrate concentration conditions when *tubP-Gal4* was used (Figs. 1c and 3a). On the other hand, Akaluc/AkaLumine showed significantly stronger signal than Luc/D-luciferin in detecting luminescence from deep tissues in *Drosophila* (Fig. 3b, c). When using *Or42b-Gal4*, the Akaluc/AkaLumine signal was superior to that of Luc/D-luciferin even under the high concentration of 10 mM substrate conditions, although the Luc/D-luciferin signal did not seem to peak even at 10 mM D-luciferin (Fig. 3d). Problems have also been reported with the administration of AkaLumine-HCl. The first is that administration of AkaLumine-HCl generates non-specific signals. A significantly higher non-specific signal was detected with AkaLumine-HCl administration than with D-luciferin, even in naïve mice[33]. In contrast, no non-specific signal was detected when AkaLumine-HCl was administered to wild-type *Drosophila* (Fig. 1b). The second problem is the toxicity of AkaLumine-HCl. AkaLumine-HCl administration has been reported to be toxic to the skin and heart, possibly due to the acidity of its solution[33,34]. We used a method of mixing AkaLumine-HCl with fly food for administering it orally and observed no noticeable toxicity in both short- and long-term administration (Fig. 2). However, it should be noted that we have not completely ruled out the possibility that further long-term administration may have adverse effects on flies. In fact, it has been reported that the pH in the food fed to flies affects the lifespan and other aspects of the flies[35]. In summary, the problems with Akaluc/AkaLumine shown in mammalian studies are generally not a problem in *Drosophila*, which further emphasizes the usefulness of Akaluc/AkaLumine in *Drosophila* research.

In our experiments, the Akaluc fusion protein was found to be a non-negligible limitation of Akaluc/AkaLumine utilization in *Drosophila*. Experiments with $brp^{Akaluc-KI}$ flies did not show an increase in Brp levels due to sleep deprivation as reported in previous studies[22] (Supplementary Fig. 4b–e). This failure to increase Brp levels is most likely due to the fusion of Akaluc to Brp, and the fusion of Akaluc may affect Brp translation, transport, localization, and degradation. The sequence of Akaluc is more than twice that of GFP (Akaluc: 1,653 bp), and the fusion of this relatively large molecule may cause the adverse effects described above. Therefore, although our experiments with *Xbp1-Akaluc* showed similar results to previous reports, it should be noted that the effects of Akaluc fusion on splicing, translation, etc. of *xbp1* cannot be completely ruled out.

NanoLuc is a luciferase derived from deep-sea shrimp, has a smaller molecular weight than firefly luciferase, and NanoLuc/furimazine releases 100 times more luminescence than Fluc/D-luciferin[36]. Due to its small molecular weight, NanoLuc may be able to solve the above problems of Akaluc fusion to proteins. Although we did not compare the NanoLuc-based luminescence

system with Akaluc/AkaLumine in this study, it is possible that Akaluc/AkaLumine is superior in detecting signals from deep tissue because NanoLuc/furimazine emits at a blue wavelength. However, in recent years, many new furimazine analogs with red-shifted emission wavelengths or with enhanced aqueous solubility have been developed, and evaluation of these furimazine analogs and NanoLuc-based systems is being conducted vigorously in mice[37–39]. Advances in comparison and evaluation of Akaluc/AkaLumine and NanoLuc-based systems in *Drosophila* will improve bioluminescence-based analysis in *Drosophila*.

Overall, Akaluc/AkaLumine is a much more powerful tool for gene and protein expression analysis in deep tissues and small numbers of cells in *Drosophila* than the conventional method Luc/D-luciferin. In addition, Akaluc/AkaLumine can be used to monitor flies non-invasively. Therefore, it is possible to measure flies under free-moving conditions and perform continuous analyses using the same individuals. Akaluc/AkaLumine enables the analysis of in vivo phenomena that are difficult using conventional methods.

## Materials and methods

***Drosophila* stock and culturing condition**. The following fly strains were used: *w^1118*, *tubP-Gal4*, *elav-Gal4*, *OK107-Gal4*, *Or42b-Gal4*, *Or85a-Gal4*, *mex1-Gal4*, and *UAS-Luc* (Bloomington Stock Center, Indiana University, USA), *UAS-Venus-Akaluc*, *UAS-Xbp1-Venus-Akaluc*, *brp^Akaluc-KI*, *IBIN^Akaluc-KI*, *IBINp-Venus-Akaluc*, and *IBINp-Venus-Akaluc-PEST* (this study). In Supplementary Fig. 1, *strip-myc* flies were used[40]. These flies were reared on normal fly food and in a 25 °C, 60% humidity, 12 h light and 12 h dark incubator. Adult females (3–5 days old) were used for all experiments and measurements, except those shown in Supplementary Figs. 1 and 4.

**Preparation of gene constructs and generation of fly strains**. To prepare the *UAS-Venus-Akaluc* construct, the sequence of *Venus-Akaluc* was amplified by PCR using *pcDNA3 Venus-Akaluc* (RIKEN DNA BANK, RDB_15781) as a template and inserted into the *pUAST-attB* vector digested with *Eco*R I and *Xho* I using Ligation high Ver.2 (TOYOBO, Osaka, Japan).

To generate the *IBIN^Akaluc-KI* strain, sense and antisense oligos were designed using flyCRISPR to prepare gRNA expression vectors. The annealed oligo sets were inserted into *U6b* plasmid digested by *Bbs* I using Ligation high Ver.2. The following primers were used: sense, CTTCGGTATCCTCCCCAG TCCTCG; antisense, AAACCGAGGACTGGGGAGGATACC (gRNA target-1); sense, CTTCGATCACGAAACTCAACCCAC; and antisense, AAACGTGGGTTGAGTTTCGTGATC (gRNA target-2).

The donor plasmid for *IBIN^Akaluc-KI* generation was constructed in two steps. First, a 1 kbp upstream sequence of *IBIN* containing the 5′ UTR was amplified by PCR from *Drosophila* genomic DNA (5′ homology arm (HA)), and the *Venus-Akaluc-SV40 late terminator* sequence was amplified by PCR from the *UAS-Venus-Akaluc* plasmid. These two fragments were inserted into the *Not* I-digested *pHD-DsRed-attP* plasmid using the NEBuilder HiFi DNA Assembly Master Mix (New England BioLabs, Ipswich, MA, USA)(*5′ HA-Venus-Akaluc-DsRed*). Next, 1 kbp downstream of the 3′ UTR of *IBIN* was amplified by PCR from *Drosophila* genomic DNA, and this fragment was inserted into *Spe* I-digested *5′ HA-Venus-Akaluc-DsRed* using NEBuilder.

In the preparation of the *IBINp-Venus-Akaluc* construct, a 319 bp region between *IBIN* and a gene 5′ upstream (*CG30109*) was amplified from fly genomic DNA as the promoter of *IBIN*. The *UAS* sequence was removed from the *UAS-Venus-Akaluc*

plasmid using *Hind* III and *Eco*R I, and the *IBIN* promoter fragment was inserted into the cut vector using NEBuilder.

To prepare the *IBINp-Venus-Akaluc-PEST* construct, the sequences of *Venus-Akaluc* were amplified by PCR except for the stop codon at the Akaluc 3′ end from *UAS-Venus-Akaluc* plasmid. Next, the *Venus-Akaluc* sequence of *IBINp-Venus-Akaluc* was cleaved with *Eco*R I and *Not* I, and the PCR-amplified *Venus-Akaluc* and artificially synthesized *PEST* sequences were introduced using NEBuilder.

To generate the *brp^Akaluc-KI* strain, sense, and antisense oligos were designed using flyCRISPR to prepare gRNA expression vectors. The annealed oligo sets were inserted into *U6b* plasmid digested with *Bbs* I using Ligation high Ver.2. The following oligo sets were used: sense, CTTCGCAATTGGTACAAATGTCGC and antisense, AAACGCGACATTTGTACCAATTGC (gRNA target-1); sense, CTTCGACAGAAGGACTCTCGAGTT, and antisense, AAACAACTCGAGAGTCCTTCTGTC (gRNA target-2).

To construct the donor plasmid to generate the *brp^Akaluc-KI* strain, first, a portion of approximately 2.8 kbp of *Brp* sequence (2 R:9536078..9538942) was amplified by PCR from *Drosophila* genomic DNA and inserted into the *Eco*R I-digested *pUC19* vector. To prevent re-cleavage by Cas9, mutations were introduced into the PAM of the target sequences of the two gRNAs of this *Brp* sequence. Next, the PCR-amplified *Akaluc* fragment from the *UAS-Venus-Akaluc* plasmid was introduced into the *Aar* I-digested *pHD-ScarlessDsRed* plasmid. From this plasmid, the *Akaluc-DsRed* fragment was PCR-amplified and then constructed using NEBuilder with linearly stranded *pUC19-Brp*, which was PCR-amplified, excluding the stop codon of *Brp*.

To generate the *UAS-Xbp1-Venus-Akaluc* construct, fragments of *5′ UTR-Xbp1*, *Venus-Akaluc* and *3′ UTR (Xbp1)* were inserted into the pUAST-attB vector. The *5′ UTR-Xbp1* fragment was amplified by PCR from *UAS-Xbp1-EGFP* fragment as a template which was PCR amplified from *UAS-Xbp1-EGFP* (HG indicator)[26] fly genomic DNA. The *Venus-Akaluc* fragment was amplified by PCR from *UAS-Venus-Akaluc* plasmid. The *3′ UTR* fragment of *Xbp1* was amplified from the *pUAST-Xbp1-EGFP* (HG indicator) plasmid. These three fragments were inserted into *Eco*R I-digested *pUAST-attB* vector using NEBuilder.

Gene constructs were injected into *Drosophila* embryos by BestGene Inc. (Chino Hills, CA, USA) and GenetiVision Corporation (Houston, TX, USA).

**AkaLumine and D-luciferin stocks**. AkaLumine-HCl (FUJI-FILM) was diluted to 60 mM in PBS and stored at −80 °C under light shielding. D-luciferin (FUJIFILM) was diluted to 100 mM in PBS and stored at −80 °C, shielded from light.

**Luminescence measurement**. To feed AkaLumine or D-luciferin to flies, normal fly food was heated and melted, and stock solutions of AkaLumine or D-luciferin were added to achieve the target concentration and administered orally. One female was placed in each well of a 24-well plate (PerkinElmer, Waltham, MA, USA) and covered with a gas-permeable seal (NIPPON Genetics, Tokyo, Japan). Fly food containing the substrate was added to the wall of the well so that the flies could ingest the substrate during measurements. Luminescence was measured using a highly sensitive bioluminescence measuring device (CHURITSU, Japan, CL-24W) at each predetermined time. Luminescence measurements using this device counted all the wavelengths that could be detected. To stop substrate administration, the flies were incubated on ice for 5 min and then transferred to a plate containing normal fly food. In all luminescence measurements, except in Fig. 4, the photometric time

was 1 s, and the photons were counted using the top and bottom detectors.

**Gene expression analysis using real-time PCR.** In Fig. 2a, b, total RNA was extracted from 10 flies (*w1118*) administered AkaLumine (1.0 mM) for 24 h. In Fig. 5d, the dorsal thorax of *IBINp-Venus-Akaluc* virgin females was pierced with a glass needle tipped with *E. coli* (DH5α), and flies were collected to extract total RNA immediately after piercing, 1, 2, 8, 12, 18, and 24 h after piercing. The uninfected group was pierced with a glass needle without *E. coli* and samples were collected simultaneously. Flies were collected 0.5 h before they were pierced with the glass needle for total RNA extraction (not pierced with the glass needle). At each time point, total RNA was extracted from five flies.

After each sample was frozen in liquid nitrogen, total RNA was extracted by homogenization in TRIzol™ Reagent (Thermo Fisher Scientific, Waltham, MA, USA) using a BioMasher™ II (Nippi, Japan). A PrimeScript™ RT reagent kit with gDNA Eraser (TaKaRa Bio, Kusatsu, Shiga, Japan) was used to remove genomic DNA and synthesize cDNA from total RNA (500 ng). For Real-time PCR, Luna Universal qPCR Master Mix (New England BioLabs) and CFX Connect (BIORAD, Hercules, CA, USA) were used. To calculate the relative expression levels of each gene, we used *Gapdh2* (Fig. 2a), *Actin 5 C* (Fig. 2b), and *RpL32* (Fig. 5d) as internal controls. Real-time PCR was performed using the following primers:

*Gapdh2*: 5′-CCCATAGAAAGCGCTCAAAA-3′ and 5′-CCAATCTTCGACATGGTTAACTT-3′,

*Actin 5 C*: 5′-TCCAGTCATTCCTTTCAAACC-3′ and 5′-CAGCAACTTCTTCGTCACACA-3′,

*GstD1*: 5′-GAGTTCCTGAACACCTTCCTG-3′ and 5′-ATTGGCGTACTTGCTGATCTC-3′,

*Bip*: 5′-GCTATTGCCTACGGTCTGGA-3′ and 5′-CATCACACGCTGATCGAAGT-3′,

*Atg8a*: 5′-TTCATTGCAATCATGAAGTTCC-3′ and 5′-GGGAGCCTTCTCGACGAT-3′,

*Hid*: 5′-TCGACCTCCACGCCGTTATC-3′ and 5′-CCTCATGATCGCTCTGGTACTC-3′,

*RpL32*: 5′-GGTTACGGATCGAACAAGCG-3′ and 5′- TTCTGCATGAGCAGGACCTC-3′,

*IBIN*: 5′- CAACTGCTGCCAATCCTCG-3′ and 5′- GCCTGGGATCGTAGTCACTT-3′.

**Investigation of the effects of AkaLumine administration on *Drosophila* development.** Eighty *w1118* females and twenty males were placed in empty bottles covered with a grape juice plate. The bottles were inverted (so that the grape juice plate was on the bottom) and allowed the flies to lay eggs for 2 h in a 25 °C incubator. To prepare the grape juice plate, grape juice (45 ml), distilled water (27 ml), glucose (4.3 g), sucrose (2.1 g), Agar (1.6 g), and 5 N NaOH (600 μl) were mixed and autoclaved. Phosphoric acid (32 μl) and propionic acid (320 μl) were added to this mixture and dispensed into 35 mm Petri dishes. Eggs were gently picked using a platinum loop, 20 eggs were placed on fly food containing AkaLumine (1.0 mM) and control fly food. For the control fly food, an equal volume of PBS was added instead of AkaLumine. Fly eggs were developed in a 25 °C, 60% humidity, 12 h light and 12 h dark incubator, and the number of flies that eventually became pupae and adults was counted.

**Bioluminescence imaging.** Akaluc flies were administered AkaLumine (1.0 mM), and Luc flies were administered D-luciferin (5.0 mM) for 24 h before luminescence measurements were taken. The control group was fed food without substrate. These flies

were attached with glue to 40 mm diameter plastic petri dishes (upper side: control group, lower side: substrate-treated group). A NightOWL II LB983 (BERTHOLD, Bad Wildbad, Germany) and IndiGo2 software were used for bioluminescence imaging and image processing. Luminescence measurements were taken under conditions of 20 s exposure time and measurement wavelengths of 650 nm for Akaluc flies and 570 nm for Luc flies. The same luminescence count pixel range was used for *tubP-Gal4 > UAS-Venus-Akaluc* and *tubP-Gal4 > UAS-Luc* image processing and *elav-Gal4 > UAS-Venus-Akaluc* and *elav-Gal4 > UAS-Luc* image processing.

**Luminescence measurements using bacteria-infected flies.** Female virgin flies were used to prevent bacterial infections caused by mating. *IBINAkaluc-KI*, *IBINp-Venus-Akaluc*, and *IBINp-Venus-Akaluc-PEST* were administered AkaLumine (1.0 mM) prior to the start of luminescence measurements and were measured for 24 h while being fed AkaLumine. The flies were then collected from the plate under cold anesthesia and the dorsal thorax was pierced with a glass needle tipped with *E. coli* (DH5α) under $CO_2$ anesthesia. The uninfected group was pierced in the same manner using a glass needle without *E. coli*. The flies were returned to the plate, and luminescence was measured again for 24 h. Only the data from flies that were still alive after the end of the measurements were used for analysis.

**Sleep deprivation.** *brpAkaluc-KI* flies were collected in vials containing food with AkaLumine (1.0 mM). The sleep deprivation group was deprived of sleep for 24 h by vibrating the vials at random times at a rate of 5 s min$^{-1}$ using a vortex mixer. The control group was fed AkaLumine without sleep deprivation. These flies were then transferred to a 24-well plate and luminescence measurements were performed for 6 h while administering AkaLumine (1.0 mM).

**Western blotting.** Adult fly brains were dissected in 1x PBS. 7–10 dissected brains were homogenized in 50 μl 1x PBS and sonicated after addition of 50 μl 2x Sample buffer (2%SDS、80 mM Tris-HCl pH6.8、15 % glycerol、0.0025 % Brilliant Blue FCF) followed by centrifugation (12,000 × *g* for 10 min at 4 °C). The supernatant was boiled for 3 min after addition of DTT the final concentration of which is 40 mM and loaded into 5% SDS-PAGE followed by immunoblot. The following primary antibodies were used: nc82 (mouse anti-BRP, Developmental Studies Hybridoma Bank, 1:1000), and DM1A (mouse anti-alpha-tubulin, Cedarlane Laboratories Ltd, CLT9002, 1:1000). Peroxidase AffiniPure Donkey Anti-Mouse IgG (Jackson ImmunoResearch, 715-035-151, 1:5000) was used as the secondary antibody. Pierce™ ECL Plus Western Blotting Substrate (thermos scientific, 32132) was used for the signal detection. Band intensities were quantified with ImageLab.

**Heat shock.** Flies administered AkaLumine (1.0 mM) for 24 h were placed in a 37 °C incubator and heat shocked for 1 h. Flies were allowed to ingest AkaLumine (1.0 mM) food during heat shock. Luminescence measurements were started immediately after the heat shock. Measurements were performed for 5 h while AkaLumine (1.0 mM) was administered to the flies.

**Statistics and reproducibility.** Prism9 software (GraphPad) was used to analyze the data. Data are shown as means ± s.e.m (error bars). Statistical tests, statistical significance, and number of trials are described in the legends of each graph. Luminescence measurement and imaging experiments were performed at least twice

to make sure that similar results could be reproduced. Experiments for real-time PCR, Brp experiments and measurement of pupariation and eclosion rates were performed once. Statistical significance was set at $p < 0.05$.

**Reporting summary**. Further information on research design is available in the Nature Portfolio Reporting Summary linked to this article.

## Data availability
Numerical source data supporting the results of this study are provided as 'Supplementary Data 1'. The Akaluc transgenic flies are available through the KYOTO Drosophila Stock Center. The plasmid vectors prepared in this study can also be provided via the RIKEN BRC DNA BANK.

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

## Acknowledgements
We are grateful to RIKEN DNA BANK for *pcDNA3 Venus-Akaluc* and Dr. Hyung Don Ryoo (NYU Grossman School of Medicine, New York, USA) for *UAS-Xbp1-EGFP. HG* plasmid and fly strains, and the Bloomington *Drosophila* Stock Center for the fly stocks used in this study. We are also grateful to Prof. Masaki Mizunuma (Hiroshima University, Hiroshima, Japan) for the research discussions, and Ms. Satoko Okazaki and Ms. Miwako Kitamura for their technical support. We thank Chihara Laboratory members for their helpful discussions. We would like to thank Editage (www.editage.com) for English language editing. This work was supported by the Frontier Development Program for Genome Editing, Astellas Foundation for Research on Metabolic Disorders, Naito Foundation, Takeda Science Foundation, JSPS KAKENHI (JP21H02479 and JP21K18236) to T.C., JSPS KAKENHI (JP20K15903) to M.O., and a JSPS Research Fellowship (JP21J22346) to A.I.

## Author contributions
A.I. and T.C. conceived the study. A.I. performed all experiments except the *IBINp-Venus-Akaluc*-related and *brp*-related experiments. N.M. constructed *IBINp-Venus-Akaluc* plasmid and performed the related experiments. Y.U. constructed *brp^{Akaluc-KI}* plasmid and performed the related experiments. T.C. and M.O. supervised the study. The manuscript was written by A.I. and T.C., with input from all authors.

## Competing interests
The authors declare no competing interests.
