## [Peer Review File · Communications Biology]

Reviewers' comments:

Reviewer #1 (Remarks to the Author):

The authors employ the Akaluc/Akalumine system to follow cells by bioluminescence (BLI) in *Drosophila* in vivo. This represents an important improvement over the standard Luciferase/Luciferin system to achieve more sensitive BLI. Overall, this is a study of highest technical quality and will have a strong impact on the *Drosophila* field. The elegant studies provide proof-of-concept that Akaluc can be used GAL4/UAS system and in knock-in alleles. There are also very useful concepts by delivering the substrate through the food, and by adding a PEST domain to Akaluc to improve temporal dynamics.

I have only minor comments and the paper is otherwise ready for publication.

Minor comments:

1. The authors may cite and compare and contrast their results to other recent studies that have utilized Akaluc recently (e.g., do a PUBMED search for "Akaluc"). For example, Bozec et al. (PMID: 33241215) as example of a study that uses Akaluc as in vivo probe for mammalian cancer cells, or Amadeo et al., PMID: 34313817
2. I appreciate that it is very difficult to determine exactly how much more sensitive Akaluc is, but it would be helpful for the readers to give some numerical estimates, such as ~10-100 fold more sensitive etc.
3. The figure 4 feels out of order, it should be presented earlier in the manuscript.
4. If reagents generated in this paper can be shared through resource centers, it should be indicated.
5. page 2, line 48: should say "was developed through" instead of "develops"

Reviewer #2 (Remarks to the Author):

COMMSBIO-23-1383 review 06-2023

This manuscript describes the application of the engineered Akaluc/AkaLumine system to continuous gene expression reporting in live *Drosophila* and comparison of the performance of this system with the traditional firefly luciferase/d-luciferin reporter in several experimental paradigms.

Critically, the authors investigated the efficacy of oral administration of the AkaLumine substrate and evaluated its potential toxicity. These toxicity studies are convincing and the reported results will be very useful for the research community. The authors discuss the downsides of oral administration in terms of the stability of the signal and limitations of the applicability (e.g. not usable in pupae), which are also very useful points of discussion that will be highly appreciated by other researchers. It would be helpful to include an analysis of the variability in Akaluc signal in individual flies over time with this delivery method in order to estimate how quantitative the bioluminescent readout of biological signal changes can actually be.

The authors convincingly demonstrate that Akaluc/AkaLumine is a more quantitative reporter for deep tissues and small cell populations compared with Luc/d-luciferin. Bioluminescence imaging was markedly more achievable with Akaluc/AkaLumine, which is an exciting development that hopefully will also translate to larger animal models. The spatial resolution of the bioluminescence imaging should be more clearly indicated.

The utility of Akaluc/AkaLumine for quantitatively monitoring expression from specific promoters was investigated using the promoter of an infection-responsive gene. The authors performed appropriate control experiments and determined that the half-life of Akaluc limited the fidelity of expression reporting, and nicely demonstrate that adding a degradation tag to the protein improves the achievable time resolution. Additional quantification and discussion of the time resolution in comparison with qPCR would be helpful in this section, as it is unclear exactly how much the performance improved with addition of the PEST sequence to Akaluc. From the data presented, it is clear that the Akaluc-PEST signal decreases to near baseline between 8 and 24h in this experiment, but since there are no qPCR time points between 8 and 24h, it is unclear whether the addition of the PEST sequence has allowed the Akaluc signal to temporally match the qPCR signal. One or two additional time points (e.g. 12h, 18h) for qPCR would be very good to include here if possible.

Akaluc was next used to monitor endogenous protein levels using a protein whose levels are increased by sleep deprivation, leading to an interesting observation of possible sex differences in this response. This observation was not discussed further in the manuscript. Without further discussion of the observed discrepancy between male and female flies in this assay, there is some question as to whether this experiment truly validated the use of Akaluc/AkaLumine for endogenous protein level monitoring, especially since the signal increase observed in male flies was also quite modest. A second example protein fusion with more conclusive results would be helpful here if it is not possible for the authors to more clearly explain the full results of the Brp experiment.

Additionally, the authors do not include any experiments to demonstrate that the Brp-Akaluc fusion is able to traffic and localize in an equivalent way to unlabeled endogenous Brp-Akaluc. Additional experiments would be helpful here – preferably imaging of antibody-labeled Brp alongside Brp-Akaluc to demonstrate equivalence. Considering the large size of Akaluc, it is not unreasonable to expect potential misfolding, disruption of protein-protein interactions, and mis-trafficking for endogenous proteins fused to it, and so it is critical to include data characterizing the fusion protein's behavior and functionality in this context. Alternatively, the authors should discuss the limitations of the data as presented to make it clear that this experiment alone does not fully demonstrate successful use of Akaluc for this specific purpose. In the interest of swift publication, including such discussion would be an acceptable way to address these concerns.

Akaluc was used to monitor alternative splicing related to ER stress using the transcript for Xbp1. This experiment is among the most convincing in terms of temporal resolution of bioluminescent reporter matching the time scale of the biological response. The same concerns as voiced above (regarding the equivalence of fusion proteins to their unlabeled endogenous counterparts) apply to this experiment, though in this case the biological event being reported is the alternative/unconventional splicing rather than protein expression level. Because of this distinction, I do not feel that additional characterization of the fusion proteins should be strictly required for this experiment, though some brief discussion of potential limitations (see above) would be beneficial.

In summary, this manuscript presents some very compelling data on the use of the Akaluc/AkaLumine system as a reporter in intact *Drosophila*. Experiments on the oral delivery of AkaLumine, use of destabilized Akaluc as a continuous expression reporter, and use of Akaluc as a reporter of alternative splicing are all presented well and are convincing, but could benefit from some minor clarifications and additions of data if possible, as noted above. Experiments to demonstrate the use of Akaluc to report endogenous protein levels were less than convincing and lacked supporting experiments to demonstrate the functional equivalence of the untagged endogenous Brp protein and its Akaluc fusion. Overall, this is a well-written report that presents data that will be highly useful to the research community. I recommend acceptance of this manuscript for publication, with the request that the authors present additional data where available or readily obtainable (additional qPCR time points would be especially welcome) and perform minor revisions to highlight limitations of the Brp-Akaluc experiment.

Reviewer #3 (Remarks to the Author):

This manuscript describes the use of the AkaLumine/AkaLuc bioluminescent system for whole animal imaging of *Drosophila*. This could be a useful tool for those who want additional means of interrogating this useful research animal. However, there are flaws to this manuscript that should be addressed prior to publication.

There are no references to *Drosophila* bioluminescence in the manuscript, and the entire introduction refers to issues with mammalian bioluminescent imaging. This is most obvious with the continual reference to issues with "deep tissue imaging" – how deep is the nervous system in *Drosophila* and how light absorbing is the tissue? This should be addressed specifically in the manuscript either through reference or experimentation.

There is no comparison to any other bioluminescent system beyond D-luciferin/Luc2. This is a well-used system but far from the only system that has been shown to work well in bioluminescence imaging. Explanation for this as the comparator system and why other comparators were not chosen is completely missing.

A third major issue is that the comparison to luciferin/Luc2 is flawed. It is well reported in the literature that luciferin has a lower K_m for Luc2 than essentially all aminoluciferins such as AkaLumine have for their respective luciferases. This is well represented when the signal is still increasing linearly at 5mM for luciferin and plateauing at 1mM for AkaLumine. But there is no true reason for not adding more luciferin – it is cheap, readily available, non-toxic and known to be effective at high concentration. These two systems should be compared at equimolar concentrations.

A final significant concern is that the Brp protein example does not recapitulate the behavior predicted by the literature, with no explanation or troubleshooting performed. This example should not be included without more understanding of the unexpected behavior, and certainly does not reassure that endogenous tagging is effective and unperturbing to the Gene of Interest.

My recommendation for this manuscript would be to re-focus on the performance of

AkaLuc/AkaLumine in *Drosophila*, with references to previous bioluminescence in *Drosophila*.

Comparisons to luciferin/Luc2 should be done at saturating substrate levels, and other bioluminescent systems should be considered.

We greatly thank all the Reviewers for their constructive and critical comments. We have responded to the Reviewers' comments individually below. We have also revised the manuscript based on each comment, highlighting the revisions in the revised manuscript.

Reviewer #1 (Remarks to the Author):

The authors employ the Akaluc/Akalumine system to follow cells by bioluminescence (BLI) in *Drosophila* in vivo. This represents an important improvement over the standard Luciferase/Luciferin system to achieve more sensitive BLI. Overall, this is a study of highest technical quality and will have a strong impact on the *Drosophila* field. The elegant studies provide proof-of-concept that Akaluc can be used GAL4/UAS system and in knock-in alleles. There are also very useful concepts by delivering the substrate through the food, and by adding a PEST domain to Akaluc to improve temporal dynamics.

I have only minor comments and the paper is otherwise ready for publication.

Minor comments:

1. The authors may cite and compare and contrast their results to other recent studies that have utilized Akaluc recently (e.g., do a PUBMED search for "Akaluc"). For example, Bozec et al. (PMID: 33241215) as example of a study that uses Akaluc as in vivo probe for mammalian cancer cells, or Amadeo et al., PMID: 34313817

We thank the Reviewer #1 for the constructive suggestion. We cited the two papers you mentioned as examples in the "Discussion" section and compared these to our results (line 280 to 303). A study by Bozec et al. (PMID: 33241215) described improved signal detection from the nervous system with the use of Akaluc/AkaLumine, as well as the usefulness of Akaluc for monitoring the small number of glioma cells transplanted and their expansion (*Neurooncol Adv.* 2(1), vdaa134, 2020). Consistent with this study, our results also show that Akaluc/AkaLumine improves sensitivity in detecting signals from the nervous system as well as from a small number of cells, such as olfactory receptor neurons (Figure 3b-e). In addition, a study by Amadeo et al. (PMID: 34313817) showed that Luc/D-luciferin has a stronger signal than Akaluc/AkaLumine when the substrate concentration is increased (*Eur J Nucl Med Mol Imaging* 49, 796–808, 2022). Consistent with this report, when the Luciferase gene was expressed in the whole body of *Drosophila*, Luc/D-luciferin released a stronger signal than Akaluc/AkaLumine under high substrate concentrations (Figure 1c and 3a). In contrast, Akaluc/AkaLumine was significantly superior to Luc/D-luciferin in detecting signals from the *Drosophila* nervous system (Figure 3b-e). These results indicate that the use of

Akaluc/AkaLumine is useful in *Drosophila* research, especially in the nervous system and in small numbers of cells.

Furthermore, we compared our results with other Akaluc/AkaLumine studies, especially with previous studies that mentioned the drawbacks of AkaLumine-HCl. In several studies including Amadeo et al. (*Eur J Nucl Med Mol Imaging* **49**, 796–808, 2022) and Nakayama et al. (*Int J Mol Sci* **21**, 1–8, 2020), administration of AkaLumine-HCl to mice has been reported to produce non-specific signals and cytotoxicity. In contrast, in our study with *Drosophila*, non-specific signals were not detected when wild-type flies were administered AkaLumine-HCl (Figure 1b), and no noticeable toxicity was observed with both short- and long-term administration of AkaLumine-HCl (Figure 2). These results further reinforce the utility of Akaluc/AkaLumine in *Drosophila*. We are grateful to Reviewer #1 because your constructive advice has helped us to emphasize the importance of our research.

2. I appreciate that it is very difficult to determine exactly how much more sensitive Akaluc is, but it would be helpful for the readers to give some numerical estimates, such as ~10-100 fold more sensitive etc.

Thanks for your advice. In our experiments, the most considerable signal difference was observed when comparing Akaluc/AkaLumine and Luc/D-luciferin in the nervous system. The use of Akaluc/AkaLumine resulted in an approximately 5-fold increase in detected luminescence compared to Luc/D-luciferin (Figure 3b; *elav>Akaluc* + 1.0 mM AkaLumine vs. *elav>Luc* + 1.0 mM D-luciferin). We have added the statements in the “Discussion” section (line 280 to 283) regarding this numerical estimate.

3. The figure 4 feels out of order, it should be presented earlier in the manuscript.

Thank you for the suggestion. We have reconsidered the order of Figures. In this study, the main analysis is based on luminescence measurements. Since the bioluminescence imaging experiment in Figure 4 was intended to reinforce the results of the Akaluc and Luc luminescence level comparison in Figure 3, we believe that the position of Figure 4 should not be changed. We are sorry that we could not fulfill your request despite your constructive suggestion.

4. If reagents generated in this paper can be shared through resource centers, it should be indicated.

Thank you for your suggestion. We have generated several Akaluc strains of *Drosophila melanogaster* in this study. We plan to donate these flies to the KYOTO Drosophila Stock Center; we can provide Akaluc transgenic flies to other researchers through this institution. We have added a sentence referring to this in the “Data availability” section of the manuscript (line 491 to 492).

5. page 2, line 48: should say “was developed through” instead of “develops”

Thank you for pointing it out. We have corrected it as Reviewer #1 suggested (line 48).

Reviewer #2 (Remarks to the Author):

COMMSBIO-23-1383 review 06-2023

This manuscript describes the application of the engineered Akaluc/AkaLumine system to continuous gene expression reporting in live *Drosophila* and comparison of the performance of this system with the traditional firefly luciferase/d-luciferin reporter in several experimental paradigms.

Critically, the authors investigated the efficacy of oral administration of the AkaLumine substrate and evaluated its potential toxicity. These toxicity studies are convincing and the reported results will be very useful for the research community. The authors discuss the downsides of oral administration in terms of the stability of the signal and limitations of the applicability (e.g. not usable in pupae), which are also very useful points of discussion that will be highly appreciated by other researchers.

It would be helpful to include an analysis of the variability in Akaluc signal in individual flies over time with this delivery method in order to estimate how quantitative the bioluminescent readout of biological signal changes can actually be.

Thanks for your constructive suggestion. As Reviewer #2 suggested, we show each fly's signal changes after AkaLumine administration in Supplemental Figure 2 (line 94 to 96). Although there were some differences, there was no significant signal variation among individuals. Akaluc/AkaLumine is considered to reach its peak reaction rate at lower substrate concentrations than Luc/D-luciferin (Figure 1c and 3a), and this feature contributes to less variation among individuals in oral administration.

The authors convincingly demonstrate that Akaluc/AkaLumine is a more quantitative reporter

for deep tissues and small cell populations compared with Luc/d-luciferin. Bioluminescence imaging was markedly more achievable with Akaluc/AkaLumine, which is an exciting development that hopefully will also translate to larger animal models. The spatial resolution of the bioluminescence imaging should be more clearly indicated.

Thank you for your critical comment. We added an experiment to analyze the spatial resolution of luminescence imaging with Akaluc/AkaLumine (line 157 to 162, Supplemental Figure 3). We have investigated whether it is possible to identify the shape of the gut from the luminescence signal by expressing Akaluc specifically in the gut of *Drosophila*. Unfortunately, we could not discern the shape of the gut but could detect signals only from the abdomen, where the gut is located. Thus, Akaluc/AkaLumine is challenging to image with enough spatial resolution to distinguish the shape of organs. Still, it is possible to reflect the location of body parts, such as the head or abdomen of flies.

The utility of Akaluc/AkaLumine for quantitatively monitoring expression from specific promoters was investigated using the promoter of an infection-responsive gene. The authors performed appropriate control experiments and determined that the half-life of Akaluc limited the fidelity of expression reporting, and nicely demonstrate that adding a degradation tag to the protein improves the achievable time resolution. Additional quantification and discussion of the time resolution in comparison with qPCR would be helpful in this section, as it is unclear exactly how much the performance improved with addition of the PEST sequence to Akaluc. From the data presented, it is clear that the Akaluc-PEST signal decreases to near baseline between 8 and 24h in this experiment, but since there are no qPCR time points between 8 and 24h, it is unclear whether the addition of the PEST sequence has allowed the Akaluc signal to temporally match the qPCR signal. One or two additional time points (e.g. 12h, 18h) for qPCR would be very good to include here if possible.

Thank you for the constructive suggestion. We have added two-time points (12 and 18 h after bacterial infection) for the qPCR experiments (Figure 5d), indicating the quick reduction of *IBIN* mRNA level after the peak at 8 h. This qPCR result well reflects the signal changes obtained from *IBINp-Venus-Akaluc-PEST* flies. Therefore, we conclude that the *IBINp-Venus-Akaluc-PEST* flies can quantitatively monitor the expression level of *IBIN* gene.

Akaluc was next used to monitor endogenous protein levels using a protein whose levels are increased by sleep deprivation, leading to an interesting observation of possible sex differences in this response. This observation was not discussed further in the manuscript.

Without further discussion of the observed discrepancy between male and female flies in this assay, there is some question as to whether this experiment truly validated the use of Akaluc/AkaLumine for endogenous protein level monitoring, especially since the signal increase observed in male flies was also quite modest. A second example protein fusion with more conclusive results would be helpful here if it is not possible for the authors to more clearly explain the full results of the Brp experiment.

Additionally, the authors do not include any experiments to demonstrate that the Brp-Akaluc fusion is able to traffic and localize in an equivalent way to unlabeled endogenous Brp-Akaluc. Additional experiments would be helpful here – preferably imaging of antibody-labeled Brp alongside Brp-Akaluc to demonstrate equivalence. Considering the large size of Akaluc, it is not unreasonable to expect potential misfolding, disruption of protein-protein interactions, and mis-trafficking for endogenous proteins fused to it, and so it is critical to include data characterizing the fusion protein's behavior and functionality in this context. Alternatively, the authors should discuss the limitations of the data as presented to make it clear that this experiment alone does not fully demonstrate successful use of Akaluc for this specific purpose. In the interest of swift publication, including such discussion would be an acceptable way to address these concerns.

Thank you for the constructive and critical comments. To confirm the reproducibility of the Brp-Akaluc results, we again performed the sleep deprivation experiments with *brp^{Akaluc-KI}*. Then, unexpectedly, we could not obtain an increased signal from sleep-deprived *brp^{Akaluc-KI}* flies in both males and females (Supplemental Figure 4b, c). To investigate the cause of these inconsistent and unexpected results, we first quantified Brp-Akaluc protein levels by Western blotting and found that Brp-Akaluc protein was not increased in sleep-deprived *brp^{Akaluc-KI}* flies (Supplemental Figure 4d, e). We next suspected that the sleep deprivation technique (mechanical stress) we used might be problematic; hence, we performed a similar experiment with wild-type flies and observed a marked increase in Brp protein levels due to sleep deprivation (Supplemental Figure 4f). Therefore, we concluded that our method of sleep deprivation was working successfully.

As Reviewer #2 pointed out, the protein folding, transport, and subcellular localization of Brp-Akaluc in the *brp^{Akaluc-KI}* may be affected. Indeed, we noticed that the *brp^{Akaluc-KI}* strain is homozygous lethal (all the experiments with *brp^{Akaluc-KI}* were performed in heterozygous *brp^{Akaluc-KI/+}*). Since we do not have an Akaluc-specific antibody, we cannot analyze the localization patterns of Brp-Akaluc protein. From these results, unfortunately, we think the *brp^{Akaluc-KI}* strain is unsuitable for detecting Brp protein level. However, we believe these (negative) results can be helpful to give caution for the Akaluc fusion protein. We have

described these points in the “Results (line 199 to 205)” and “Discussion (line 304 to 310)” sections and alerted the reader to these problems. We are grateful to Reviewer #2 because this comment has helped us realize the limitations of using the Akaluc fusion protein.

Akaluc was used to monitor alternative splicing related to ER stress using the transcript for Xbp1. This experiment is among the most convincing in terms of temporal resolution of bioluminescent reporter matching the time scale of the biological response. The same concerns as voiced above (regarding the equivalence of fusion proteins to their unlabeled endogenous counterparts) apply to this experiment, though in this case the biological event being reported is the alternative/unconventional splicing rather than protein expression level. Because of this distinction, I do not feel that additional characterization of the fusion proteins should be strictly required for this experiment, though some brief discussion of potential limitations (see above) would be beneficial.

Thank you for your helpful advice. In the paper that developed an improved Xbp1-EGFP, the authors discussed the possibilities of generating an artifactual EGFP signal that is not derived from unconventional splicing of *xbp1* (*Cell Stress and Chaperones* 18, 307-319, 2013). The first possible mechanism is the generation of abnormal EGFP fusion proteins due to unexpected translation starting at the ATG codon, which is not the initiation codon of unconventionally spliced *xbp1*. Another possible mechanism is proteolytic digestion at the junction of the XBP1 and EGFP proteins. It is possible that a similar phenomenon may produce an unexpected Akaluc fusion protein or Akaluc separated from XBP1. However, our temporal luminescence measurements from Xbp1-Akaluc showed that the rate of signal decrease was more similar to the half-life of XBP1 than to the half-life of the Akaluc protein (Figure 6c and 6e). In addition, luminescence at the basal level without heat stress was significantly lower than after heat stress. These results suggest that the unconventional splicing-independent Akaluc signal is low or negligible.

Although the results obtained from Xbp1-Akaluc well fit previous reports (*Nature Medicine* 17, 1251–1260, 2011. and *Metabolism* 105, 154046, 2020), it is still possible that the fusion of Akaluc gene to Xbp1 gene may affect the splicing and translation level of Xbp1 gene. We mentioned this in the “Discussion (line 310 to 312)” section.

In summary, this manuscript presents some very compelling data on the use of the Akaluc/AkaLumine system as a reporter in intact *Drosophila*. Experiments on the oral delivery of AkaLumine, use of destabilized Akaluc as a continuous expression reporter, and use of Akaluc as a reporter of alternative splicing are all presented well and are convincing, but

could benefit from some minor clarifications and additions of data if possible, as noted above. Experiments to demonstrate the use of Akaluc to report endogenous protein levels were less than convincing and lacked supporting experiments to demonstrate the functional equivalence of the untagged endogenous Brp protein and its Akaluc fusion. Overall, this is a well-written report that presents data that will be highly useful to the research community. I recommend acceptance of this manuscript for publication, with the request that the authors present additional data where available or readily obtainable (additional qPCR time points would be especially welcome) and perform minor revisions to highlight limitations of the Brp-Akaluc experiment.

We thank Reviewer #2 for the critical comments and recommend our manuscript to be published after the revision. We hope the Reviewer #2 is satisfied with our responses.

Reviewer #3 (Remarks to the Author):

This manuscript describes the use of the AkaLumine/AkaLuc bioluminescent system for whole animal imaging of *Drosophila*. This could be a useful tool for those who want additional means of interrogating this useful research animal. However, there are flaws to this manuscript that should be addressed prior to publication.

There are no references to *Drosophila* bioluminescence in the manuscript, and the entire introduction refers to issues with mammalian bioluminescent imaging. This is most obvious with the continual reference to issues with “deep tissue imaging” – how deep is the nervous system in *Drosophila* and how light absorbing is the tissue? This should be addressed specifically in the manuscript either through reference or experimentation.

We thank Reviewer #3 for these comments. The position of the fly nervous system is not deeper than mammals such as mice (The head of *Drosophila* is about 1 mm thick). However, it has been reported that light absorption or scattering by pigments in the eyes and cuticle of *Drosophila* causes hindrance in the fluorescent observation of tissues (*Nat Commun* 9: 4731, 2018). Our experiments have shown that the fluorescent signals detected from the fly body are very different depending on the presence or absence of the *white* gene required for eye pigment synthesis, and the *yellow* gene required for melanin synthesis in the cuticle (Supplemental Figure 1). Therefore, applying Akaluc/AkaLumine, which emits more tissue-permeable luminescence than Luc/D-luciferin, could improve signal detection sensitivity from deep tissues, such as the nervous system, in *Drosophila*. This was mentioned in the

“Introduction (line 52 to 56)” section.

There is no comparison to any other bioluminescent system beyond D-luciferin/Luc2. This is a well-used system but far from the only system that has been shown to work well in bioluminescence imaging. Explanation for this as the comparator system and why other comparators were not chosen is completely missing.

We thank Reviewer #3 for the critical comment. In *Drosophila* research, the Fluc/D-luciferin luminescence system has generally been used to date. When we searched the *Drosophila* database (based on FlyBase data through August 30, 2023) for the number of research papers used for each luciferase, Fluc (*P. pyralis* Luc) was used predominantly (The *UAS-Luc* strain used in this study is also derived from *P. pyralis*). Thus, we determined a comparison between the Luc/D-luciferin luminescence system, which is the most widely used in *Drosophila* research, and Akaluc/AkaLumine in this study. In the “Introduction (line 61)” and “Discussion (line 282)” sections, we have added a sentence referring to the fact that Luc/D-luciferin has been commonly used in *Drosophila* research. In addition, we also mentioned the NanoLuc, which has a smaller molecular weight, may be helpful to solve the problems with the Akaluc fusion protein in the “Discussion (line 313 to 322)”.

A third major issue is that the comparison to luciferin/Luc2 is flawed. It is well reported in the literature that luciferin has a lower K_m for Luc2 than essentially all aminoluciferins such as AkaLumine have for their respective luciferases. This is well represented when the signal is still increasing linearly at 5mM for luciferin and plateauing at 1mM for AkaLumine. But there is no true reason for not adding more luciferin – it is cheap, readily available, non-toxic and known to be effective at high concentration. These two systems should be compared at equimolar concentrations.

We thank Reviewer #3 for the constructive suggestion. As Reviewer #3 suggested, we applied a higher concentration of D-Luciferin (10 mM) to *tubP-Gal4>UAS-Luc* flies. We found that the signal is comparable with that from 5.0 mM D-Luciferin (Figure 3a), suggesting that signal has already reached a plateau at a substrate concentration of 5.0 mM. We changed

the description “the signal did not appear to peak even at 5.0 mM D-luciferin” to “the signal peaked at 5.0 mM D-luciferin” in the manuscript (line 242).

We also applied 10 mM substrate concentration to flies expressing Akaluc or Luc in the nervous system (Figure 3b-d). When we expressed Luc in all neurons (*elav-Gal4*) or mushroom body neurons (*OK107-Gal4*), the signals from Luc-expressing flies were not different between 5.0 mM and 10 mM. On the other hand, the higher concentration of D-Luciferin (10 mM) provided more signals than 5.0 mM D-Luciferin when Luc was expressed in a small number of neurons (*Or42b-Gal4>UAS-Luc*).

A final significant concern is that the Brp protein example does not recapitulate the behavior predicted by the literature, with no explanation or troubleshooting performed. This example should not be included without more understanding of the unexpected behavior, and certainly does not reassure that endogenous tagging is effective and unperturbing to the Gene of Interest.

(As we received the same criticism from Reviewer #2, too, we use the same response here.) We thank Reviewer #3 for this critical comment. To confirm the reproducibility of the Brp-Akaluc results, we again performed the sleep deprivation experiments with *brp^{Akaluc-KI}*. Then, unexpectedly, we could not obtain an increased signal from sleep-deprived *brp^{Akaluc-KI}* flies in both males and females (Supplemental Figure 4b, c). To investigate the cause of these inconsistent and unexpected results, we first quantified Brp-Akaluc protein levels by Western blotting and found that Brp-Akaluc protein was not increased in sleep-deprived *brp^{Akaluc-KI}* flies (Supplemental Figure 4d, e). We next suspected that the sleep deprivation technique (mechanical stress) we used might be problematic; hence, we performed a similar experiment with wild-type flies and observed a marked increase in Brp protein levels due to sleep deprivation (Supplemental Figure 4f). Therefore, we concluded that our method of sleep deprivation was working successfully.

As Reviewer #3 pointed out, the function of endogenous protein could be affected by Akaluc tagging (Brp-Akaluc in the *brp^{Akaluc-KI}*). Indeed, we noticed that the *brp^{Akaluc-KI}* strain is homozygous lethal (all the experiments with *brp^{Akaluc-KI}* were performed in heterozygous *brp^{Akaluc-KI/+}*). Since we do not have an Akaluc-specific antibody, we cannot analyze the localization patterns of Brp-Akaluc protein. From these results, unfortunately, we think the *brp^{Akaluc-KI}* strain is unsuitable for detecting Brp protein level. However, we believe these (negative) results can be helpful to give caution for the Akaluc fusion protein. We have described these points in the “Results (line 199 to 205)” and “Discussion (line 304 to 310)” sections and alerted the reader to these problems. We are grateful to Reviewer #2 because

this comment has helped us realize the limitations of using the AkaLuc fusion protein.

My recommendation for this manuscript would be to re-focus on the performance of AkaLuc/AkaLumine in *Drosophila*, with references to previous bioluminescence in *Drosophila*. Comparisons to luciferin/Luc2 should be done at saturating substrate levels, and other bioluminescent systems should be considered.

We hope the Reviewer #3 is satisfied with our responses.

REVIEWERS' COMMENTS:

Reviewer #2 (Remarks to the Author):

The authors have satisfactorily addressed all of the points from my original review, and this manuscript should be accepted for publication.

Reviewer #3 (Remarks to the Author):

I commend the authors on doing additional experimentation, particularly with regard to the Brp fusion, and adjusting the manuscript accordingly. This manuscript will be an asset to any researcher wanting to use AkaLuc/AkaLumine in Drosophila. I recommend publication.